# An anti-inflammatory activation sequence governs macrophage transcriptional dynamics during tissue injury in zebrafish

Nicolas Denans [1] ✉, Nhung T. T. Tran[1], Madeleine E. Swall [1], Daniel C. Diaz[1,2], Jillian Blanck[1] & Tatjana Piotrowski [1] ✉

Macrophages are essential for tissue repair and regeneration. Yet, the molecular programs, as well as the timing of their activation during and after tissue injury are poorly defined. Using a high spatio-temporal resolution single cell analysis of macrophages coupled with live imaging after sensory hair cell death in zebrafish, we find that the same population of macrophages transitions through a sequence of three major anti-inflammatory activation states. Macrophages first show a signature of glucocorticoid activation, then IL-10 signaling and finally the induction of oxidative phosphorylation by IL-4/ Polyamine signaling. Importantly, loss-of-function of glucocorticoid and IL-10 signaling shows that each step of the sequence is independently activated. Lastly, we show that IL-10 and IL-4 signaling act synergistically to promote synaptogenesis between hair cells and efferent neurons during regeneration. Our results show that macrophages, in addition to a switch from M1 to M2, sequentially and independently transition though three anti-inflammatory pathways in vivo during tissue injury in a regenerating organ.

Innate immune cells, and more particularly macrophages, are essential during vertebrate embryonic development, tissue repair and regeneration[1,2]. Ablating macrophages during, or after injury of major organs in regenerating species blocks the regeneration process[3–6]. Yet, the sequence of signals that controls their activity and their dynamics at a high spatio-temporal resolution are still poorly defined.

Macrophages exist in various molecular activation states. They are commonly classified as pro-inflammatory (M1) or anti-inflammatory (M2) depending on the type of cytokines/signals they get activated with[7]. However, this classification that stems from a comparison with lymphocyte T helper (Th) cell activation[7] is an oversimplification. For one, zebrafish larvae do not have functional Th cells yet, making the nomenclature irrelevant. Furthermore, M1 macrophages can be activated by several pro-inflammatory signals such as Interleukin-1, Interferon-gamma, lipopolysaccharide (LPS), while M2 are triggered by anti-inflammatory Glucocorticoids, Interleukin-10, Interleukin-4/13 and more[8]. In addition, each of these signals triggers distinct

downstream gene regulatory networks depending on the context and can have distinct functions. Nevertheless, a switch from M1 to M2 has been observed in different contexts of tissue injury (pathogen-induced-, sterile- or mixed injury).

Two models have been proposed and observed after tissue injury: a "phenotypic switch", where the same population switches from pro- to anti-inflammatory[9–14] or the "independent recruitment" of pro- and anti-inflammatory populations sequentially[15–17]. Which model is at play during tissue regeneration is still poorly documented. A recent study in a model of tailfin regeneration in zebrafish demonstrated that the same macrophages can switch from pro-inflammatory (tnfa + ) to anti-inflammatory (cxcr4b + ) during the course of regeneration[18] favoring the phenotypic switch model. However, whether cells can sequentially transition between several pro- or anti-inflammatory states during tissue injury is understudied mainly due to the lack of high spatio-temporal resolution analyses. Indeed, the majority of injury paradigms studied thus far occurs over many days, making it challenging to

[1]Stowers Institute for Medical Research, 1000 east 50th street, Kansas City, MO 64110, USA. [2]Present address: Parse Biosciences, 201 Elliott Ave W, Suite 290, Seattle, WA 98119, USA. ✉e-mail: ndenans@stowers.org; pio@stowers.org

observe fast transitions between different macrophage activation states. Determining both the timing of the transitions as well as the genetic programs triggered by each activation state during tissue regeneration will be invaluable to design targeted immunomodulatory therapies.

Here we take advantage of the rapid sensory hair cell (HC) regeneration that occurs in the zebrafish lateral line[19,20] to identify the molecular "What" (signals and genetic program) and "When" (sequence of activation) that control macrophage activity and dynamics during injury of a regenerating organ in zebrafish. Using a combination of high-resolution microscopy and scRNAseq, we show that a population of macrophages is sequentially and independently activated by three major anti-inflammatory pathways. Finally, a compound mutant analysis demonstrates a synergistic role for IL-10 and IL-4 signaling in synaptogenesis after hair cell injury.

## Results

### The same population of effector macrophages invades and leaves neuromasts within a five-hour window

Tissue injury triggers an inflammatory response that involves the recruitment of macrophages to the injury site. These macrophages will be responsible for clearing cellular debris, remodeling the extracellular matrix, and in some cases, provide signals to the tissue stem cells to start the repair process[21]. A recent study showed that macrophages rapidly invade the neuromast and phagocytose dead HCs[22]. We designated this population 'effector macrophages'[23]. It is not clear if effectors represent a single, or several macrophage populations. A recent study of muscle regeneration in zebrafish showed that two populations of macrophages were dynamically regulated after injury[24]. One population reacted rapidly to tissue injury and left the damaged area within twenty-four hours, while a second population stayed in contact with the muscle stem cells for a longer period. Thus, the first step in understanding how effector macrophages are molecularly regulated is to describe their dynamic after HC death. To follow both macrophages and HCs over time we study *Tg(mpeg:GFP)* and *Tg(she:lckmScarletI)* transgenic larvae in which macrophages and neuromast cells are labeled, respectively (Fig. 1a). Neuromast HCs are superficially located all along the larval body (Fig. 1a, Supplementary Movie 1). Treatment with neomycin for 30 min rapidly kills HCs by caspase-independent cell death (Supplementary Movie 2)[25]. Macrophages start phagocytosing dead HCs as early as 15 min after the first cells die (Supplementary Movie 2–3)[22]. To describe the dynamics of effector macrophages from the time they enter the neuromast to when they leave during HC regeneration, we performed a macrophage recruitment assay. We treated the larvae with neomycin for 30 min and imaged both macrophages and neuromasts 1 hour (1H), 3H, 5H, and 7H after treatment (Fig. 1a, b, Supplementary Movie 4). Quantification of the number of macrophages surrounding and inside the neuromasts shows a rapid recruitment of effector macrophages with a peak at 1H after neomycin (Fig. 1c, d)[22]. Subsequently, cells progressively return to homeostatic levels, which they reach at 7H after neomycin treatment. Thus, effector macrophages interact with neuromasts within a five-hour window, coinciding with the appearance of the first regenerated HCs[26].

To decipher where effector macrophages originate from and if they represent a specific population, we performed time-lapse recordings of macrophages and neuromasts during neomycin treatment and homeostasis in a large area of the larval trunk (Supplementary Movie 5). Quantification of the location of macrophages prior to HC death shows that macrophages that will become effectors are located closer to the neuromast than macrophages that will become non-effectors (71 µm ± 14 µm vs 175 µm ± 15 µm, respectively) (Fig. 1e). In contrast to non-effectors, effector macrophages show a rapid, directional migration toward the neuromasts in response to HC death (Fig. 1f). Interestingly, non-effector cells show an increase in non-

directed cell velocity after neomycin treatment suggesting that HC death initially triggers a global injury response (Fig. 1g).

To assess if a single or multiple populations of effector macrophages are recruited to neuromasts during the five-hour window, we specifically photoconverted effector cells within neuromasts 1H after neomycin and quantified the ratio of photoconverted vs non-photoconverted macrophages 3H and 5H after HC death (Fig. 1h). Photoconverted cells represent an average of 98% ± 2% and 100% of the macrophages inside neuromasts at 3H and 5H after HC death, respectively, demonstrating that a single population of effector macrophages are recruited all at once during the five-hour window (Fig. 1i).

### High temporal resolution scRNA-seq identifies a population of effector macrophages

To molecularly characterize all macrophage populations, identify the effector population and characterize their activation sequence and underlying molecular program over the course of HC death, we performed a scRNA-seq time course of *Tg(mpeg:GFP)* transgenic larvae. We dissociated 600 5dpf larvae each during homeostasis, and 1H, 3H and 5H after a 30 min neomycin treatment and FACsorted GFP + cells for 10x Chromium genomics scRNA-seq (Fig. 2a and Supplementary Fig. 1a–c). To determine the earliest activation of macrophages we also collected GFP + cells immediately after a 15 min neomycin treatment (Fig. 2a). We downsampled the number of cells per time point to fourteen thousand to avoid a bias based on over-representation of a specific time point (Supplementary Fig. 2a). We subsequently integrated the five time points using Seurat and performed UMAP dimensional reduction (Fig. 2b). The resulting seventy thousand cell atlas of all *mpeg*:GFP cells of 5 dpf larvae show that *mpeg*:GFP, in addition to macrophages, also labels several other immune cell types, some of which have also been recently observed in 4dpf larvae[27] and adult fish[28,29]. The expression of *mpeg1.1* by these non-macrophage immune cell types is confirmed in feature and violin plots (Supplementary Fig. 1d-e). We characterized these immune cell populations based on the expression of marker genes (Fig. 2b, Supplementary Fig 2b-c and Supplementary Data 1). We identified dendritic-like cells (DC-like) based on their expression of the master regulator *flt3*[30], as well as *spock3* and *hepacam2*. Previously described antigen-presenting metaphocytes[31] are marked by *cldnh, epcam,* and *prox1a*. Two clusters of natural-killer like cells express their master regulator *eomesa*[32] (NK-like) and *gata3*[33] (NK-like2). A neutrophil population is labeled by *mpx* and *lyz*[34]; an unidentified population expresses skin (*rbp4, sparc*) and collagen markers (*col1a2*), and two unidentified small clusters are labeled by *kng1* and *acta3b*, respectively. Our analysis implicated an unexpected heterogeneity of the macrophage population consisting of eight clusters that cluster closely together (Fig. 2b, Supplementary Fig. 2b-c and Supplementary Data 1). Proliferating macrophages form a cluster characterized by *pcna, mki67* and *tubb2b*. Another cluster that is heavily influenced by ribosome genes we called the 'translation' cluster. Cells in the small 'stat1b' cluster show an Interferon signaling response signature (*stat1b, isg15, cxcl20*) that is also induced in response to an endemic picornavirus in zebrafish facilities[35]. Another cluster is marked by genes classically upregulated in response to bacteria[36,37] that we named 'irg1/acod1' (*irg1/acod1, hamp, mxc*). In addition, we identified a cluster that does not show unique markers but is broadly labeled by *f13a1b, junba,* and *btg1* (called 'f13a1b'); a cluster that expresses markers of potentially immature microglia, such as *mcamb, apoeb*[38], and *apoc1* (called 'mcamb'); and two clusters that represent unidentified macrophage states or populations expressing *runx3, cxcl19, cxcl8a* (called 'runx3') and *tspan10, slc43a3b, illr1* (called 'tspan10'), respectively.

To characterize macrophage clusters that show transcriptional changes during the neomycin time course, we performed differential gene expression for each cluster at each time point. Quantification of the numbers of up- and down-regulated genes during the time course

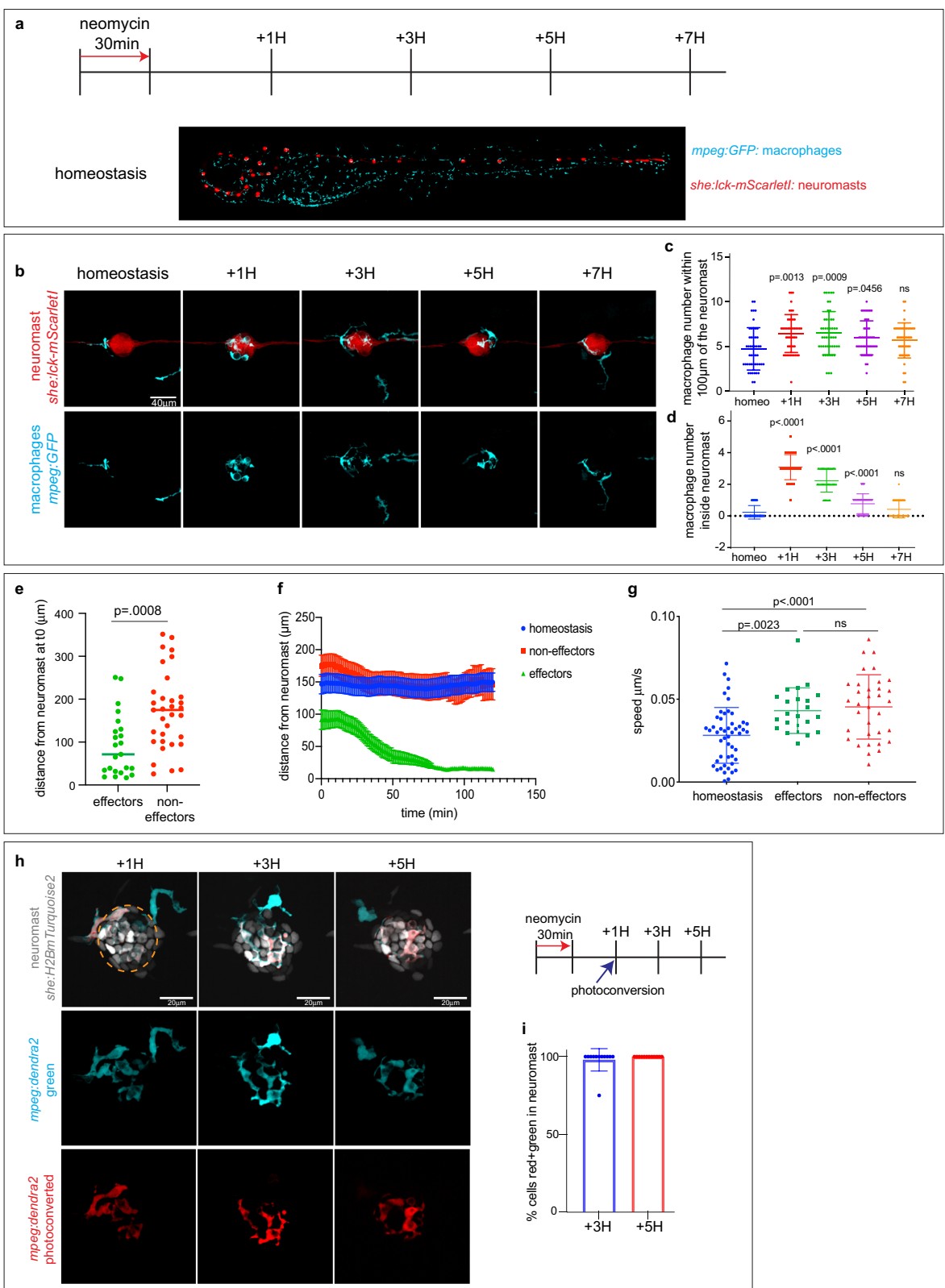

shows that several macrophage clusters respond to neomycin treatment (Fig. 2c, d, h, l and Supplementary Fig. 3a-b). To identify which macrophages enter the neuromasts and represent effector cells, we performed fluorescent in situ hybridization using the hybridization chain reaction (HCR-FISH[39]) with the macrophage cluster markers *irg1/acod1*, *f13a1b*, *mcamb*, *tspan10*, *runx3*, as well as the non-macrophage NK-like (*eomesa*) and DC-like (*hepacam2*) cluster markers as negative controls (Fig. 2d–o, Supplementary Fig. 3a–d"). HCR-FISH was performed on *mpeg:GFP* larvae 1H after neomycin, when all effector cells have migrated into the neuromasts. This screen demonstrated that 48% of the effectors are labeled with *irg1/acod1* (Fig. 2d–g), 18% with *f13a1b* (Fig. 2h–k) and 34% with *mcamb* (Fig. 2l–o). The *tspan10* marker did not label any effector cells (Supplementary Fig. 3a–a"), while we found that 3% of the cells were labeled with *runx3*

**Fig. 1 | The same population of macrophages resolves HC death-induced inflammation within a five-hour window. a** Schematics of neomycin regime and time point collections for the macrophage recruitment assay in the *Tg(she:lck-mScarletI;mpeg:GFP)* larvae. **b** Representative confocal images (projection of a 30 mm z-stack) for the macrophage recruitment assay. **c**, **d** Quantification of macrophages around (**c**) or inside (**d**) the neuromast. Each dot represents the number of macrophages per neuromast (3 neuromasts per larvae, 16 larvae per condition and 3 biological replicates). A 2-way ANOVA followed by a Tukey multiple comparison tests has been used to determine statistical significance. *P*-values represent a post-hoc (Tukey) test from each condition relative to homeostasis. **e** Quantification of the distance from the neuromast prior to HC death of effector and non-effector macrophages. Each dot represents a macrophage (8 larvae per condition and 3 biological replicates). *P*-value represents a two-tailed Student's *t*-test. **f** Quantification of the distance from the neuromast over-time after HC death

of effector and non-effector macrophages (8 larvae per condition and 3 biological replicates). **g** Quantification of the macrophage velocity of effectors and non-effectors macrophages. Each dot represents a macrophage (8 larvae per condition and 3 biological replicates). A 2-way ANOVA followed by a Tukey multiple comparison test has been used to determine statistical significance. *P*-values represent a post-hoc (Tukey) test between each condition. **h** Representative confocal images (projection of a 30 mm z-stack) for the macrophage photoconversion assay. Cells photoconverted (red) were inside the neuromast at the time of photoconversion while non-photoconverted cells (cyan) were not. Orange dotted line represents the photoconverted area. **i** Quantification of the ratio of photoconverted effector macrophages over time. Each dot represents the percentage of macrophages per neuromast (1 neuromast per larvae, 12 larvae per condition and 3 biological replicates). For all graphs, data are represented as mean ± SD except for (**f**) which is represented as mean ± SEM.

(Supplementary Fig. 3b-b"). As expected, no effector cells were labeled with *eomesa* (NK-like cells) (Supplementary Fig. 3c-c") or *hepacam2* (DC-like cells) (Supplementary Fig. 3d-d"). Additionally, we performed a macrophage recruitment assay with two newly generated transgenic reporter lines that drive the expression of a red fluorescent protein under the promoter of *irg1/acod1* and *stat1b*. We functionally validated that *Tg(−5.6irg1*:lck-mScarletI/*acry*:mScarletI) larvae respond to LPS injection and that this line is thus a faithful reporter of the 'irg1/acod1' cluster. Similarly, our *Tg(−9stat1b*:lck-mScarletI/*acry*:mScarletI) reporter is activated by *Ifnphi1* injection (Supplementary Fig. 4a–d). These experiments confirmed that 'irg1/acod1' cells are indeed effector macrophages, whereas 'stat1b' cells do not enter neuromasts (Supplementary Fig. 4e–h). Altogether we conclude that the effector population is composed of cells belonging to the 'irg1/acod1', 'f13a1b' and 'mcamb' clusters and focused our analysis on these cells as effector macrophages.

## An anti-inflammatory activation sequence in macrophages during HC death

To identify the core molecular programs driving the effector macrophages activation states at each time point, we compared the differentially expressed genes of the three clusters and analyzed the genes that were shared (Supplementary Data 2). We detected that at the fifteen minutes neomycin time point the Glucocorticoid (GR) pathway targets (*arl5c, jdp2b, dusp1, klf9, tsc22d3, nfkbiab, sgk1, fkbp5*)[40,41] are strongly and transiently upregulated (Fig. 3a, Supplementary Fig. 5a, Supplementary Data 2). HCR-FISH for *dusp1* demonstrated that this activation of the GR pathway is systemic (Fig. 3b). Likewise, *interleukin10 receptor alpha* (*il10ra*) is also strongly upregulated (Fig. 3a, Supplementary Fig. 5a, Supplementary Data 2). We confirmed the expression of *il10ra* in 75% of effector cells by HCR-FISH (Fig. 3c-d). Gene ontology and pathway analyses using Metascape[42] showed enrichment for genes involved in the regulation of apoptosis (*gadd45ab, gadd45bb, xiap, gbp, btg2, ddit3 and mcl1a*) and GR pathway activation (Supplementary Data 2) within the upregulated genes, while 'oxidative phosphorylation' is the most enriched term within the downregulated genes at the 15 min time point (Supplementary Fig. 5b, Supplementary Data 2). Other markers of anti-inflammatory macrophages that have not been linked to GR activation, such as *cxcr4b, irf2, lpn1, mknk2* and *ets2*[43–47] are also strongly upregulated (Supplementary Fig. 4A, Supplementary Data 2). Of note, *ncf1* and *nrros*[48,49], both negative regulators of reactive oxygen species, are strongly upregulated (Supplementary Fig. 5a, Supplementary Data 2). Unexpectedly, our stringent analysis did not show a robust upregulation of pro-inflammatory cytokines and the pro-inflammatory *il1b* is only upregulated in the 'irg1/acod1' cluster at the 15 min neomycin time point, while no transcriptional upregulation of *tnfa, il6* or *il12a* could be detected (Fig. 3a and Supplementary Fig. 6). This finding suggests that a transition from a pro-inflammatory (*il1b* +) to an anti-inflammatory (GR +) state occurs rapidly within minutes after HC death. The

immediate upregulation of the *il10ra* receptor at 15 min leads to the activation of IL-10 target genes activation at 1H after neomycin (*socs3b, ccr12b.2, pim1,* and *fgl2a*)[50–52] (Fig. 3a). HCR-FISH confirms that 80% of the effector cells express *fgl2a* (Fig. 3e, f). Other master activators of anti-inflammatory macrophage states, such as the *interleukin4 receptor il4r.1* and the rate-limiting enzymes of the polyamine pathway, *odc1* and *smox*, are also upregulated (Fig. 3a, Supplementary Fig. 5a, Supplementary Data 2). At 3H, in addition to the inhibition of pro-inflammatory cytokines by IL-4, the combination of Polyamine and IL-4 signaling induces oxidative phosphorylation in macrophages[53,54] (Fig. 3a, Supplementary Fig. 5a, Supplementary Data 2). For example, genes belonging to the mitochondria respiratory chain complexI (*ndufab1b, ndufb3, mt-nd1*), complexIII (*cycsb, uqcr10, uqcrfs1, uqcrh*), complexIV (*cox17, cox7a2a, cox7c, cox7b*) and complexV (*atp5g1, atp5g3a, atp5j, mt-atp6*) are induced (Fig. 3a, Supplementary Fig. 5a, Supplementary Data 2). Likewise, Pathway and Gene Ontology (GO) analysis shows 'Oxidative phosphorylation' as the most enriched term (Supplementary Data 2). Interestingly, *manf*, a gene required for retina repair and regeneration in fly and mouse[55] is strongly upregulated at the three-hour time point hinting toward a switch to a repair state of macrophages (Fig. 3a, Supplementary Fig. 5a). In contrast to the earlier time points, very few genes are found specifically upregulated at the 5H time point (Fig. 3a, Supplementary Fig. 5a). *grn2*, a pro-granulin growth factor regulator of the anti-inflammatory macrophage phenotype[56] is highly upregulated. Interestingly, in *grn* knock-out mice, muscle injury leads to a persistence of macrophages at the injury site suggesting that *grn* is involved in regulating macrophage dynamics[57]. Altogether, this analysis shows a linear sequence of macrophage anti-inflammatory activation immediately after HC death starting with Glucocorticoid signaling, followed by IL-10 signaling, and lastly a combination of Polyamine and IL-4 signaling, which induces oxidative phosphorylation at the transcriptional level. We report an in vivo sequence of three major anti-inflammatory activation pathways after injury. This suggests that in addition to a transition from a pro-inflammatory M1 to an anti-inflammatory M2 state, the same population of macrophages transitions through different anti-inflammatory states to potentially regulate their dynamics/function.

## Each anti-inflammatory state is independently activated

The discovery of this linear anti-inflammatory sequence of effector macrophages activation raises the interesting question if epistatic relationships between Glucocorticoid signaling and IL-10 signaling and between IL-10 signaling and IL-4/Polyamine signaling exist.

To address this question, we first inhibited Glucocorticoid signaling during HC death using the GR inhibitor RU486. HRC-FISH for the GR target *dusp1* confirmed that a 10 μM treatment with RU486 was efficient in blocking GR activation (Fig. 4a). HCR-FISH for *il10ra* at the 15 min time point, as well as for the IL-10 signaling target gene *fgl2a* at the 1H time point showed no downregulation of these genes in effector macrophages after RU486 treatment (Fig. 4b–e). This demonstrates

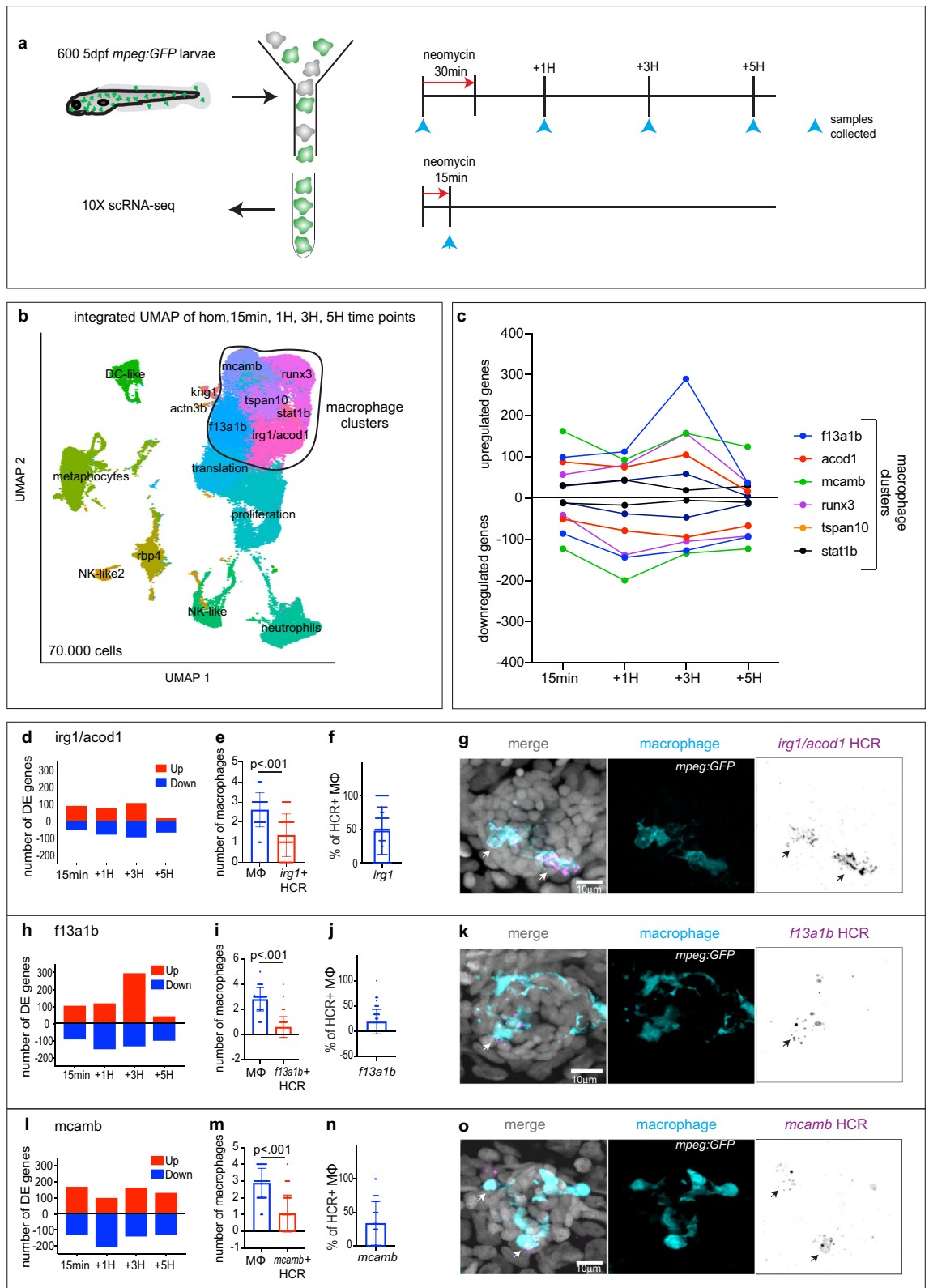

that GR activation is not required for IL-10 activation in effector macrophages.

To test for a possible epistatic relationship between IL-10 signaling and the following activation state (Polyamine + IL-4 signaling), we generated an *il10ra* mutant by deleting the promoter region, as well as the first exon using CRISPR/Cas9 (Supplementary Fig. 7a). HCR-FISH for *il10ra* in homozygous mutants confirmed the absence of transcript

(Supplementary Fig. 7b-c). We also performed HCR-FISH for the target gene *fgl2a* at the 1H time point and observed a strong downregulation in effector cells in the mutant (64%+/−7% in homozygous vs 21%+/−5% in heterozygous) (Supplementary Fig. 7d, e). Next, we performed a scRNA-seq time course after neomycin treatment in homozygous and heterozygous *il10ra* larvae (Supplementary Fig. 8a). Integration of these six datasets (sixty thousand cells) using Seurat and UMAP

**Fig. 2 | Combination of scRNA-seq and HCR-FISH identifies a population of effector macrophages during HC regeneration. a** Schematics of neomycin regime and time points collection for scRNA-seq. **b** Integrated UMAP of the five time points (1 experiment per time point). Cluster names are labeled on the UMAP. Macrophage clusters are circled. **c** Quantification of genes differentially up- and downregulated at each time point within the macrophages clusters. **d, h, l** Quantification of genes differentially up- and downregulated at each time point for (**d**) the 'irg1/acod1', (**h**) 'f13a1b' and (**l**) 'mcamb' clusters. **e, i, m** Quantifications of effector macrophages (MΦ) that either express (**e**) irg1/acod1, (**i**) f13a1b or (**m**) mcamb by HCR at 1H. Each dot represents the number of macrophages per

neuromast (5 neuromasts per larvae, 16 larvae and 3 biological replicates). *P*-values represent non-parametric two-tailed Student's *t*-test. **f, j, n** Quantifications of the percentage of HCR + effector macrophages (MΦ) in the (**f**) irg1/acod1, (**j**) f13a1b and (**n**) mcamb clusters. Each dot represents the number of effector macrophages per neuromast (5 neuromasts per larvae, 16 larvae and 3 biological replicates). **g, k, o** Representative confocal images (projection of a 30 μm z-stack) of HCR-FISH (arrows) within the effector macrophages for (**g**) irg1/acod1, (**k**) f13a1b and (**o**) mcamb (from 5 neuromasts per larvae, 16 larvae and 3 biological replicates). For all graphs, data are represented as mean ± SD.

dimensional reduction shows that the effector clusters ('irg1/acod1', 'mcamb' and 'f13a1b') are conserved in mutant larvae (Fig. 4f-g, Supplementary Fig. 8b-c, Supplementary Data 3). Likewise, quantification of the expression levels and dynamics of *il4r.1*, and *odc1* shows that they are unaffected in the mutants compared to the siblings (Fig. 4h). The subsequent activation of genes related to oxidative phosphorylation and *manf* 3H after neomycin is also unaffected (Fig. 4h and Supplementary Fig. 9). Thus, the induction of oxidative phosphorylation by IL-4/Polyamine signaling is independent of IL-10 signaling activity. This important result shows that, while a sequential induction of anti-inflammatory pathways underlies macrophage activation during HCs regeneration, its components are independently activated (Fig. 4i). Therefore, activation of a single anti-inflammatory pathway is likely not sufficient to induce proper tissue regeneration and a sequential and independent activation of the three pathways might be required.

## IL-10 and IL-4 signaling act synergistically to promote HC synapse formation with efferent neurons during neuromast regeneration but not during homeostasis

HCs form synapses with the efferent axons responsible for modulating their response to external stimuli[58–61]. Macrophages are required for promoting/maintaining synapses between HCs and neurons in several models of cochlear HCs lesions[62–64]. To assess the role of IL-10 and/or IL-4 signaling in HC synapses formation/maintenance during HC regeneration, we generated *il10ra* (IL-10 signaling) and *stat6* (downstream effector of IL-4 signaling[65]) double heterozygous mutants. We focused on IL-10 and IL-4 signaling since these pathways are specifically activated in macrophages, while GR activation is systemic, complicating the characterization of its macrophage-specific function in HC regeneration. Incrosses between *il10ra/stat6* double heterozygous mutants allowed us to analyze WT, *il10ra-/-*, *stat6-/-* and *il10ra/stat6-/-* 5dpf larvae during homeostasis, 24H and 48H after neomycin treatment (regeneration). We quantified HC numbers, as well as the ratio of efferent synapses per HC (using an antibody against Vamp2 labelling efferent synapses). We did not observe any differences in HC numbers or numbers of efferent synapses per HC in any of the mutant genotypes during homeostasis (Fig. 5a–c). However, 24H after HC death, we observed a dramatic reduction of efferent synapses per regenerating HC in the double mutant (*il10ra/stat6 -/-*) larvae compared to WT and single homozygous mutants, while the number of HCs was not significantly affected (Fig. 5d–f). The synapse phenotype perpetuates 48H after HC death, even though the number of regenerating hair cells is the same as in sibling larvae (Fig. 5g–i). The phenotype in the double mutant demonstrates that (i) IL-10 and IL-4 pathway are required for HC synapses numbers only during regeneration but not during homeostasis and (ii) that IL-10 and IL-4 act synergistically during HC regeneration to promote formation/maintenance of HC synapses. To test if the synapse phenotype could be due to a defect in macrophage recruitment to the neuromast, we quantified macrophage dynamics during the first five hours after HC death in WT and double (*il10ra/stat6-/-*) mutants. However, this analysis did not identify any changes in macrophage recruitment dynamics (Supplementary Fig. 10a–c). Altogether, these results demonstrate that IL-10 and IL-4 are required for

HC/synapse formation/maintenance during regeneration while they are dispensable for regulating the macrophage recruitment dynamics to the neuromasts.

## Discussion

The molecular activation of macrophages regulates their dynamics and activity, and the modulation of their activation can have dramatic effects on tissue repair and regeneration[66]. Here, using a high spatio-temporal resolution analysis of macrophage activation during HC regeneration, we provide evidence that the same population of macrophages is sequentially and independently activated by three major anti-inflammatory pathways.

It is now broadly documented that macrophages adopt a pro-inflammatory activation state immediately after injury[18,27,67–69]. The role of this pro-inflammatory phase is mainly to attract additional macrophages to the injury site if the size of the injury requires it[70]. This first phase must be followed by an anti-inflammatory phase to resolve the inflammation and ensure proper tissue repair. Our analysis shows that in the lateral line, a single population of tissue-resident macrophages resolves HC death-induced inflammation, favouring the phenotypic switch model. Furthermore, our data show a short pro-inflammatory phase and a rapid transition to an anti-inflammatory state marked by the strong and systemic activation of the GR pathway. A possible explanation for this lack of strong pro-inflammatory activation of macrophages after HC death is that the tissue-resident population is sufficient to resolve inflammation and the recruitment of inflammatory macrophages is not required. This hypothesis is supported by our finding that the effector macrophages reside in immediate vicinity of the neuromasts during homeostasis and that after HC loss on average only three macrophages are detected in neuromasts.

The anti-inflammatory role of the GR pathway has been extensively documented[40,71–73]. It can both directly inhibit pro-inflammatory gene transcription by direct binding of the GR receptor to their enhancers/promoters or by tethering the pro-inflammatory activators AP-1 and Nfkb[73]. Therefore, the strong and transient activation of the GR pathway immediately as the first HCs start to die likely turns off the transcription of pro-inflammatory cytokines.

The short phase of GR activation is immediately followed by IL-10 signaling activation and a subsequent transition to a IL-4/Polyamine activation state that induces oxidative phosphorylation. A recent study in mouse Bone Marrow-Derived Macrophages (BMDM) showed that part of the IL-10 anti-inflammatory function is to inhibit glycolysis while promoting oxidative phosphorylation after treatment with LPS, which mimics a bacterial infection[74]. Our *il10ra* mutant analysis demonstrates that lack of IL-10 signaling after HC death does not affect the induction of oxidative phosphorylation. Co-stimulation of mouse BMDM with LPS and IL-10 also leads to the upregulation of the IL-4 receptor[75], whereas the loss of IL-10 signaling, via the knock-out of *il10ra*, in zebrafish does not affect the induction of *il4r.1*. These discrepancies suggest that IL-10 and IL-4 signaling are differently regulated in response to bacterial infection versus tissue injury or reflect interspecies differences.

The IL-4/OXPHOS state is characteristic for wound healing macrophages and responsible for the last step of inflammation

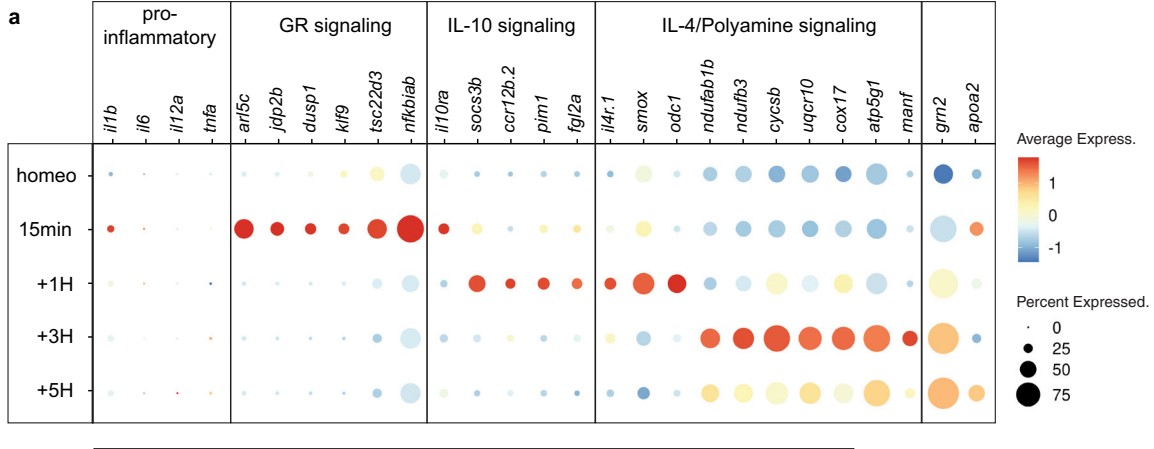

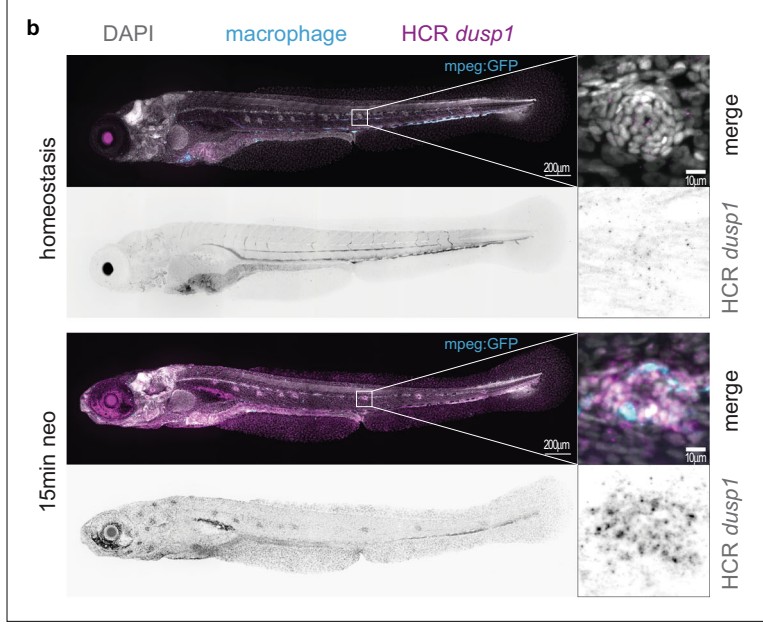

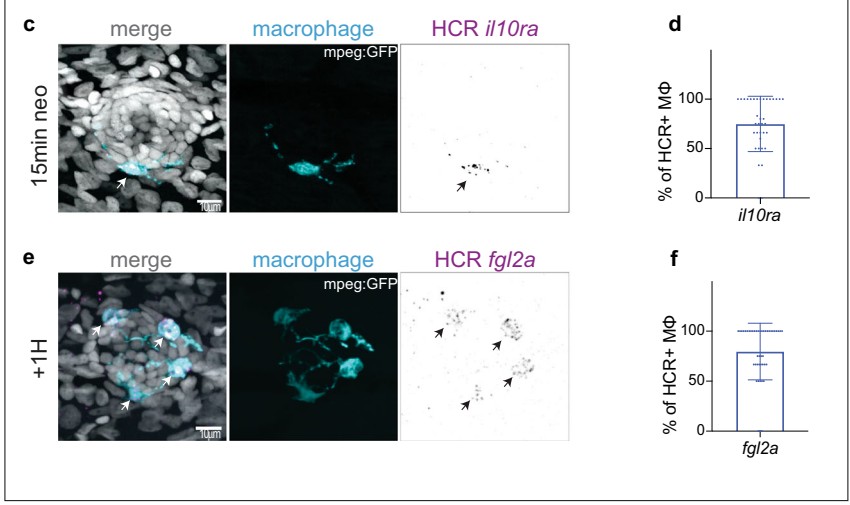

**Fig. 3 | Transcriptional dynamics show an anti-inflammatory activation sequence in effector macrophages. a** Dot-Plot of selected differentially upregulated genes for each time point of the scRNA-seq time course within the effector population. **b** Representative confocal images (Maximum projection of a 200 µm z-stack) of an HCR-FISH for *dusp1* in a 5dpf larvae (left) and zoom image of a representative neuromast (right) (from 6 whole larvae at 5dpf for homeostasis and 6 whole larvae at 5dpf after 15 min neo treatment. 2 biological replicates). **c, e** Representative confocal images (projection of a 30 µm z-stack) of HCR-FISH within the effector macrophages (arrows) for (**c**) *il10ra*, (**e**) *fgl2a*. **d, f** Quantifications of the percentage of HCR + effector macrophages (MΦ) for (**d**) *il10ra*, (**f**) *fgl2a*. (12 larvae with 3 neuromasts per larva and 3 biological replicates). For all graphs, data are represented as mean ± SD.

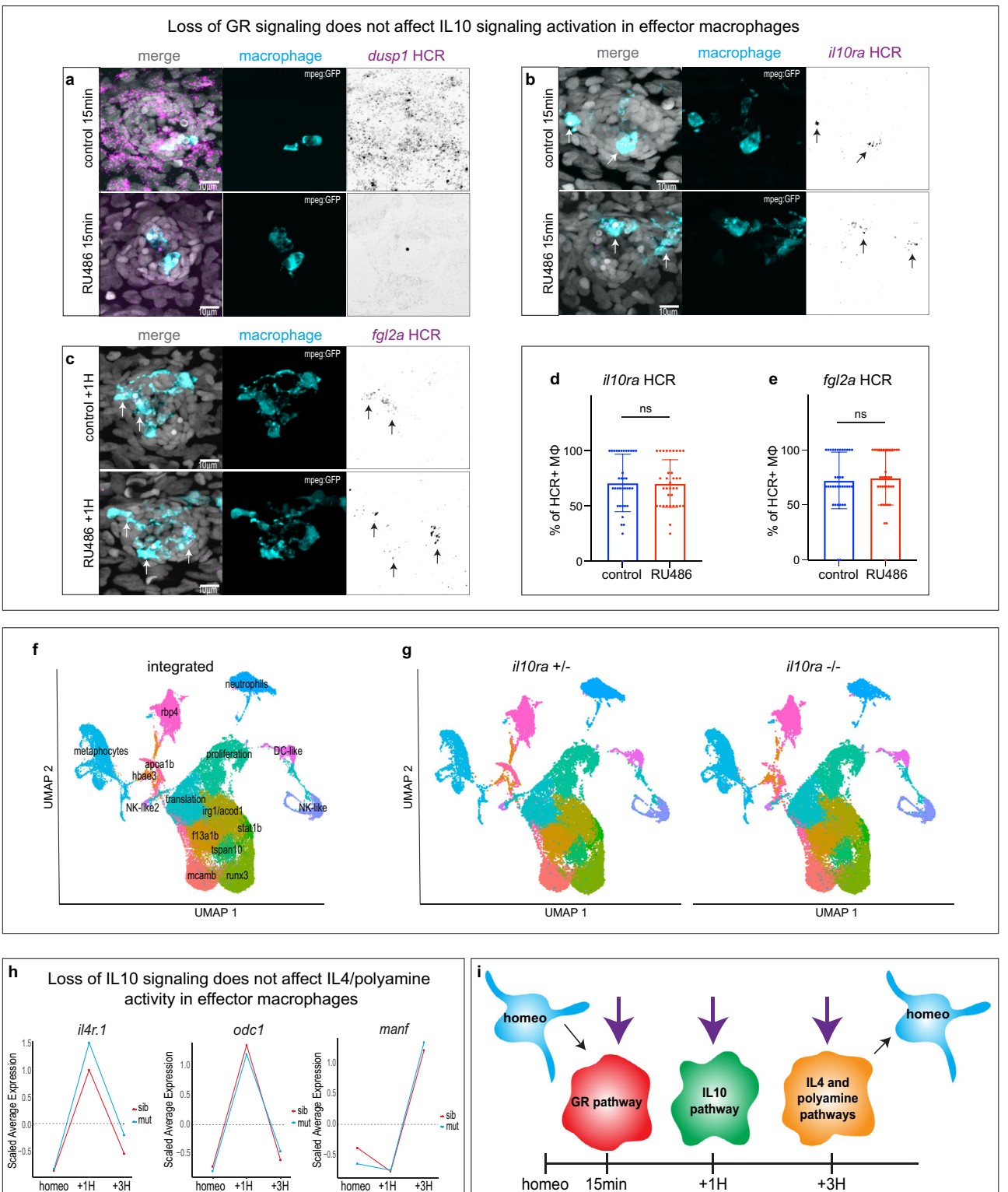

**Fig. 4 | Each anti-inflammatory state is independently activated.**
**a** Representative confocal images (maximum projection of a 30 μm z-stack) of an HCR-FISH for *dusp1* in a neuromast (10 larvae per condition, 30 neuromasts total and 3 biological replicates). **b, c** Representative confocal images (projection of a 30 μm z-stack) of HCR-FISH within the effector macrophages (arrows) for (b) *il10ra* and (c) *fgl2a* (12 larvae per conditions, 36 neuromasts total and 3 biological replicates) **d, e** Quantifications of the percentage of HCR + effector macrophages (MΦ) for (**d**) *il10ra* and (**e**) *fgl2a* (12 larvae per conditions, 36 neuromasts total and 3 biological replicates). For all graphs, data are represented as mean ± SD.

**f** Integrated UMAP of the six datasets for the *il10ra* mutant. Cluster names are labeled on the UMAP. **g** Split UMAP per condition (*il10ra ±* and *il10ra-/-*). **h** Line plots representing the average expression for each time point between the *il10ra* mutant (cyan) and the sibling (red) from the scRNA-seq integrated dataset. **i** Model of independent activation of the three anti-inflammatory pathways in effector macrophages during the HC regeneration time course. Resting macrophages are represented in cyan. Purple arrows represent the independent induction of each macrophage activation state.

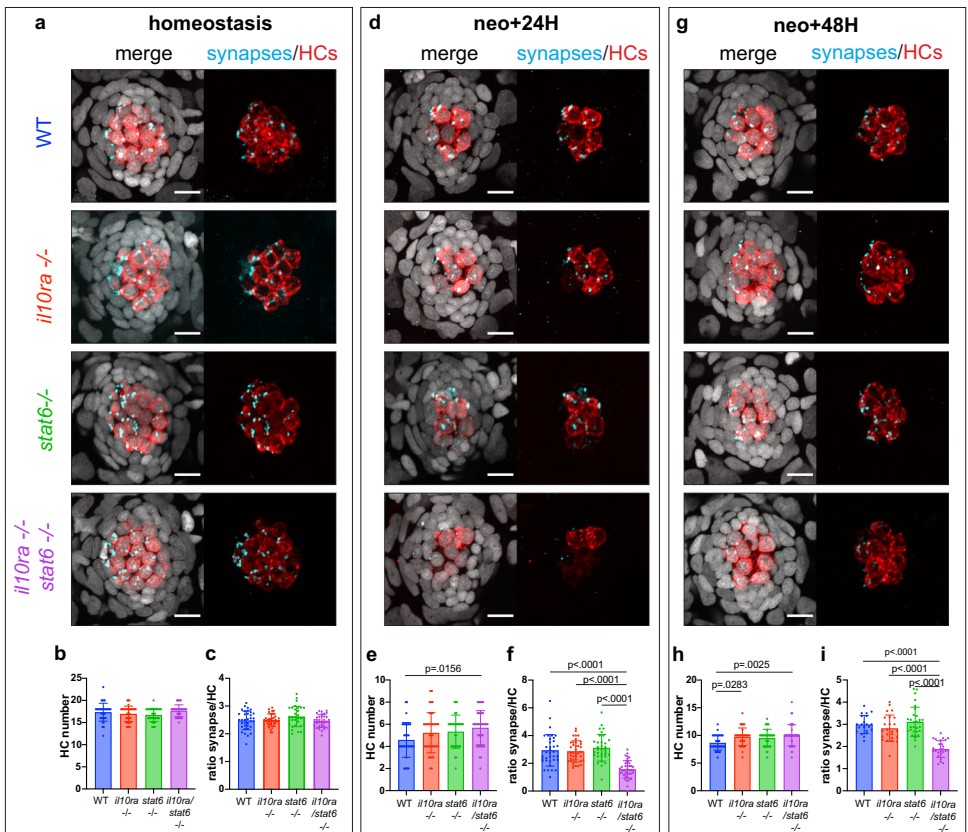

**Fig. 5 | IL-10 and IL-4 signaling act synergistically to promote HC synaptogenesis with efferent neurons during regeneration but not during homeostasis. a, d, g** Representative confocal images (maximum projection of a 10 μm z-stack) of the HC and efferent synapse regeneration assay. **b, e, h** Quantification of HC numbers during (**b**) homeostasis, (**e**) 24H after HC death and (**h**) 48H after HC death (11–14 larvae per condition, 3 neuromasts per larvae and 4 biological replicates). **c, f, i** Quantification of the number of efferent synapses per HC (ratio) during (**c**) homeostasis, (**f**) 24 h after HC death and (**i**) 48H after HC death (8–14 larvae per condition, 3 neuromasts per larvae and 4 biological replicates). A 2-way ANOVA followed by a Tukey multiple comparison test has been used to determine statistical significance. *P*-values represent a post-hoc test (Tukey) between each condition. For all graphs, data are represented as mean ± SD.

resolution[2,76–78]. In addition, the induction of the pro-repair gene *manf*, which is required for retina regeneration[55], suggests that at the 3H time point macrophages transition to a wound-healing state. Altogether, our in vivo data demonstrate the sequential and independent transition of effector macrophages through three major anti-inflammatory states.

Macrophages have been shown to be required for hair cell synaptogenesis in mice[62–64] but the signals responsible for this function remain unknown. We demonstrate that the combined loss of *il10ra* and *stat6* affect hair cell synaptogenesis after zebrafish hair cell injury but not during development. Interestingly, loss of *manf* induces a hair cell synaptogenesis phenotype in the mouse cochlea but only in the C57BL/6 J background known to have early onset hearing loss[79]. This suggests a multifactorial cause for the synaptogenesis defects and begs the question of the IL-10 signaling activity in C57BL/6 J mice.

An outstanding question remains: is the identified sequence of anti-inflammatory states that we unveiled specific to HC death or is it conserved in different organs, injury types and species? Our study is, to our knowledge, the first high spatio-temporal resolution analysis of macrophage activation after injury of a regenerating organ. Most studies focus on later time points of macrophage activation soon after injury, making it impossible to perform quantitative comparisons. Nonetheless, data from different injury paradigms, organs and species suggest that our discovery is conserved across multiple species and organs. For example, it has recently been shown that an acute heart injury in the adult zebrafish triggers glucocorticoid pathway activation in distant organs[80]. Moreover, both IL-10 and IL-4 signaling are activated in response to a muscle injury in mouse macrophages[81].

Interestingly, a recent study comparing the transcriptome of regenerating African spiny mouse (*Acomys*) versus non-regenerating *Mus Musculus* in a model of spinal cord injury highlights the activation of IL-10 and IL-4 signaling during regeneration[82]. While the major pathways (GR, IL-10, and IL-4) might be conserved, the downstream targets might be specific to different tissues. For instance, a recent study shows that macrophages provide NAMPT to the muscle stem cells after injury in zebrafish to promote tissue regeneration[24]. This might be specific to muscle injuries since the stem cells express the NAMPT receptor CCR5 but not all tissue stem cells do. Likewise, neurotrophic factors such as *manf* that we find upregulated after HC death, and which is also required for retina regeneration[55] are likely involved in sensory cell regeneration to maintain their neuronal connectivity. Thus, we believe our study should be placed in context of sterile sensory cell regeneration and will be a starting point for quantitative comparative studies manipulating one parameter at a time. Such studies that characterize not only the activation sequence of macrophages but also the downstream targets specific to different tissues, cell death types and sterility status will be invaluable to design tailored regenerative immunotherapies.

## Methods
### Zebrafish lines and husbandry
This study was conducted in accordance with the Guide of the Care and Use of Laboratory Animals of the National Institute of Health and protocols were approved by the Institutional Animal Care and Use Committees of the Stowers Institute for Medical Research (Sensory lateral line development and hair cell regeneration in zebrafish, TP

## Table 1 | Transgenic lines used in this study

| Transgenic line name | Tissue labeled | reference |
|---|---|---|
| *Tg(mpeg1.1*:EGFP)[gl22], abbreviated as *mpeg*:GFP | Immune cells | Ellet et al.[37] |
| *Tg(she*:lckmScarletI)[psi70Tg] | neuromast | This paper. Similar to[84] |
| *Tg(she*:H2BmTurquoise2)[psi72Tg] | neuromast | This paper |
| *Tg(Myo6b*:lck-mScarlet-I)[psi67Tg] | Hair cells | Peloggia et al.[84] |
| *Tg(−5.6irg1*:lck-mScarletI/*acry*:mScarletI) [psi73Tg] referred as *irg1*:lck-mScarletI | Macrophages expressing *irg1/acod1* | This paper. Similar to[89] |
| *Tg(−9stat1b*:lck-mScarletI/*acry*:mScarletI) [psi75Tg] referred as *stat1b*:lck-mScarletI | Reporter for Interferon signaling activation | This paper |
| *Tg(she*:gap43-GFP)) [psi74Tg] | neuromast | This paper |
| *Tg(mpeg1.1*:dendra2)[uwm12] referred as *mpeg*:dendra2 | Immune cells | Harvie et al.[90] |
| *il10ra* [psi71] | CRISPR KO for *il10ra* | This paper |
| *stat6 delta 10/+ (xt56)* | CRISPR KO for *stat6* | Cronan et al.[65] |

Protocol: # 2017-0176). Adult fish are maintained with a photoperiod of 14 h light and 10 h dark. Fish are fed daily twice a day with live feed (artemia) plus a dry pellet feed (Ziegler).

The following zebrafish transgenic lines were used (Table 1):

All transgenic lines were generated using the tol2 system[83].

For the *Tg(she*:lckmScarletI)[psi70Tg], *Tg(she*:H2BmTurquoise2)[psi72Tg] and *Tg(she*:gap43-GFP) [psi74Tg]: p5E-she[84] was combined with either pME-lckmScarletI or pME-H2BmTurquoise2 or pME-GAP43-GFP, with p3E-PA and pDEST-Tol2pA2 following the tol2Kit protocol[85].

For the *Tg(−5.6irg1*:lck-mScarletI/*acry*:mScarletI)[psi73Tg]: 5.6 kb directly upstream of the ATG of *irg1/acod1* were PCR amplified from genomic DNA using the following primers: Fwd 5-TCTGGTGG TAGAACGTGAGAGT-3 and Rev 3- TGTGCGAGCTCTGAATCTTGA-5. The PCR product was inserted into a vector containing Tol2 recombination sites as well as a cassette allowing for the expression of mScarletI in the fish eyes using the alpha-crystalline promoter (eye marker) using Gibson cloning. The full map will be made available upon request. 2 nl of a mix of DNA (15 ng/uL) and tol2 transposase mRNA (40 ng/uL) were injected in one cell stage of AB/TU mixed background embryos. Founders were screened by visible mScarletI expression in the F1 larvae eyes thanks to the eye marker cassette. Experiments have been performed on F2 stable generations.

For the *Tg(−9stat1b*:lck-mScarletI/*acry*:mScarletI) [psi75Tg]: 9.4 kb directly upstream of the ATG of *stat1b* were PCR amplified from genomic DNA using the following primers: Fwd 5- CAGCATA TGTCAAGTGTTCGCA-3 and Rev 3- GCAACCCGTTATGCAACCCT-5. The PCR product was inserted into a vector containing Tol2 recombination sites as well as a cassette allowing for the expression of mScarletI in the fish eyes using the alpha-crystalline promoter (eye marker) using Gibson cloning. Full map will be made available upon request. 2 nl of a mix of DNA (15 ng/uL) and tol2 transposase mRNA (40 ng/uL) were injected in one cell stage of AB/TU mixed background embryos. Founders were screened by visible mScarletI expression in the F1 larvae eyes thanks to the eye marker cassette. All experiments were performed with F2 stable generations.

The *il10ra* [psi71] mutant was generated using CRISPR/Cas9. sgRNAs were selected using CRISPRscan[86] to delete a 2 kb region from the putative promoter to the second exon in order to prevent transcription. sgRNA1: 5-GTGTTTTGGTCGGGTGTGGT-3; sgRNA2: 5-GGTACG GGGGCTTCATTGGG-3. Genotyping primers to assess for the deletion are: Fwd: 5-TGCATAACAGCTCAGCCATCTTCTC-3; Rv: 5- CATGTCA CATTCTTCCACAATGTTCC-3. sgRNA were purchased from IDT. 2 nl of a RNP mix of Cas9 protein (PNA Bio #CP01, 200 ng/uL) with 2 sgRNA (60 uM each) were injected in one cell stage of AB/TU mixed background embryos. Founders were selected by PCR genotyping. Deletion results in a 250 bp band on a 1.5% agarose gel. Experiments were performed with F2 stable generations. For experiments, homozygous were incrossed (homozygous) and compared to outcross with WT ABs (heterozygous).

### Validation of *Tg(−5.6irg1*:lck-mScarletI/*acry*:mScarletI) [psi73Tg] as a reporter for bacterial infection

Anesthetized (MS-222, up to 150 mg/L) *Tg(−5.6irg1*:lck-mScarletI/ *acry*:mScarletI) [psi73Tg]/mpeg1:GFP 4 dpf larvae were injected with either 5nL of 1 mg/mL LPS (Sigma #L2880) or PBS in the heart to mimic a bacterial infection and immediately reincubated overnight in fresh E2 medium at 28 C. Larvae were consequently fixed in 4%PFA at 5dpf overnight at 4 C. Larvae were washed 3 times in PBSTween 0.1% in incubated in DAPI (ThermoFisher #D1306; 1/2000 dilution) for 30 min followed by 3 10 min washes in PBSTween 0.1%. The trunk of each larva was imaged on a Nikon confocal spinning disk with a 20×0.95NA water immersion objective with a 2um z-step. Macrophages (mScarletI + , GFP + and both) were quantified in 3D using IMARIS 9.6.

### Validation of *Tg(−9stat1b*:lck-mScarletI/*acry*:mScarletI) [psi73Tg] as a reporter for response to Interferon

Assay was performed as described in[35]. Briefly, 1 cell stage *Tg(−9stat1b*:lck-mScarletI/*acry*:mScarletI) [psi73Tg] were injected with 50 pg of a vector coding for *InfPhi1* (PCS2- *InfPhi1* kind gift from Nels Elde) or non-injected control. Embryos were raised to 5 dpf and scored for broad expression of mScarletI or lack thereof visually under a fluorescent binocular.

### Sensory hair cell ablation

To ablate hair cells, 5dpf embryos were treated with 300 µM neomycin (Sigma-Aldrich #N6386, St Louis, MO, USA) for 30 min at 28 °C or for a 15 min pulse. Following, embryos were washed with 0.5x E2 medium (7.5 mM NaCl, 0.25 mM KCl, 0.5 mM MgSO4, 75 mM KH2PO4, 25 mM Na2HPO4, 0.5 mM CaCl2, 0.5 mg/L NaHCO3, pH = 7.4) and incubated at 28 °C until further experimental needed for further experiments.

### Embryo dissociation and FACS

600 GFP-positive 5dpf larvae were anesthetized with tricaine for ~1 min until they stopped moving (1:20 dilution of 4 g/L tricaine in 0.5x E2 medium). To dissociate the larvae, we placed them into 2 wells (300 larvae each) containing strainers (BD Falcon Cell Strainer (BD Biosciences #352350, San Jose, USA), quickly rinsed the larvae in ice-cold DPBS and added 4.5 ml cold 0.25% trypsin-EDTA (Thermo Fisher Scientific #25200056, Waltham, USA) supplemented with 1uM ActinomycinD (Sigma A1410). The larvae (300 each) were then transferred to one 5 ml polypropylene conical tube (placed on ice) with a disposable transfer pipet. The larvae were dissociated by trituration with a Pasteur pipette hooked to a pipetboy until all GFP + cells were dissociated (~7 min) on ice. Cells in suspensions were first separated from the larval bodies by filtering the suspension through a Filcons 70 µm cell strainer (BD Biosciences #340634, San Jose, CA, USA) into a 5 ml polypropylene round-bottom tube. Subsequently, the cells were centrifuged at 2000 rpm (720 x *g*) for 5 min at 4 °C. To wash off the trypsin-EDTA, we removed it, added ice-cold DPBS with 1uM ActinomycinD and

centrifuged the cells at 2000 rpm (720 x $g$) for 5 min at 4 °C. Resuspended cells in fresh ice-cold DPBS with 1uM ActinomycinD were filtered through a Filcons 70 μm cell strainer into a falcon round bottom 5 ml tube (Falcon-Corning #352063, Glendale, AZ, USA) and immediately FACSorted. Samples were sorted on a BD Influx with a 100 μm tip at 20 psi with 1X PBS, directly into chilled 90% MetOH (Sigma #34860). Prior to sorting, samples were stained with 25 uM DRAQ5 (Thermo Fisher #65-0880-92) for 5 min on ice. DRAQ5 is excited by a 647 nm laser at 100 mW with detection at 720/40 nm. GFP is excited by a 488 nm laser at 100 mW with detection at 528/28 nm. Single color controls were analyzed to confirm spectral compensation is unnecessary; the GFP + gate was set using a non-expressing sample stained with DRAQ5 as an FMO (fluorescence minus one) control.

## 10X Chromium scRNA-seq library construction

Methanol fixed cells were rehydrated with rehydration buffer (1% BSA (Sigma #A1595) and 0.5 U/μl RNase-inhibitor (Sigma #03335402001) in ice-cold DPBS (Sigma #D8537). Approximately 20.000 cells were loaded into the Chromium Single Cell Controller (10x Genomics). For library preparation, Chromium Next GEM Single Cell 3' GEM, Library Gel Bead Kit v3.1 was used. The sample concentration was measured on a Bioanalyzer (Agilent) and sequenced with Novaseq6000 Kit v3 with read length of 28 bp Read 1, 8 bp i7 index and 91 bp Read 2 (150 cycles) (Illumina).

## scRNA-seq read alignment and quantification

Raw reads were demultiplexed and aligned to version 10 of the zebrafish reference transcriptome (danRer10, Ensembl release 91) following the 10X Genomics' CellRanger (v2.1.1) pipeline for the macrophage time course dataset. Prior to the downstream QC filtering we obtained the following cell numbers: 20,676 (homeostasis), 17,187 (15 min), 29,846 (1 h), 20,540 (3 h), and 26,413 (5 h) using CellRanger's cell-association algorithm. To normalize the cell numbers across the time course, we randomly subsampled 14,000 cells from each time point. Post-filtering, the number of cells per sample were 13,937 (homeostasis), 13,908 (15 min), 13,921 (1 h), 13,886 (3 h), and 13,921 (5 h). The mean number of genes per cell per sample were 1,475 (homeostasis), 1,306 (15 min), 1,174 (1 h), 1,558 (3 h) and 1,488 (5 h).

For the *il10ra* heterozygous (also referred to as sibling) versus homozygous (also referred to as mutant) scRNA-seq dataset, raw reads were demultiplexed and aligned to version 11 of the zebrafish reference transcriptome (danRer11, Ensembl release 102) following 10X Genomics' CellRanger (v6.0.1). Prior to QC filtering, we obtained the following cell numbers for the heterozygous (sibling) samples: 14,159 (homeostasis), 12,720 (1 h), and 13,793 (3 h) using CellRanger's cell-association algorithm. We obtained the following cell numbers for the homozygous (mutant) samples: 12,004 (homeostasis), 14,183 (1 h), and 19,546 (3 h) using CellRanger's cell-association algorithm. Post-filtering, the number of cells per sibling sample were 7,201 (homeostasis), 9,517 (1 h), 10,705 (3 h) and 7,895 (homeostasis), 10,806 (1 hr), and 13,693 (3 hr) for the mutant samples. The mean number of genes per cell per sibling samples were 2,073 (homeostasis), 1,347 (1 h), and 1,560 (3 h). The mean number of genes per cells per mutant samples were 1,825 (homeostasis), 1,248 (1 h), and 1,628 (3 h). All raw data for the macrophage time course and *il10ra* sibling versus mutant samples including sorted BAM files and count matrices produced by CellRanger has been deposited in Gene Expression Omnibus (GEO) database, (#GSE209884).

## Pre-processing, quality filtering and batch integration

To distinguish zebrafish repeated gene symbols with unique Ensembl IDs, we modified the count matrices outputted from the CellRanger pipeline using a custom R (version 3.6.3) script (https://doi.org/10.5281/zenodo.6871552). Each repeated gene symbol is annotated with an asterisk followed by an incrementing number. Since the

macrophage time course and *il10ra* samples were aligned to different reference transcriptomes (Ensembl release 91 versus Ensembl release 102, respectively), we provide both gene symbol conversions in the supplemental gene lists to account for any differences between the two versions (Supplementary Data 1 and 3). For the macrophage time course dataset, low-quality cells or cells containing doublets with reads greater than 20,000, reads less than 600 and mitochondrial contamination greater than 5% for each sample were filtered from the subsequent analysis. Genes present in less than 10% of the cells were also removed from the dataset.

For the *il10ra* samples, low-quality cells or cells containing doublets with reads greater than 50,000, genes per cell less than 500, genes per cell greater than 7,500 and mitochondrial contamination greater than 5% from each sample were filtered from the subsequent analysis. Genes present in less than 10% of the cells were also removed from the dataset.

Both datasets were integrated following the standard integration pipeline outlined by the R package Seurat (v3.2.0, (Butler et al., 2018)). Here, individual temporal samples were normalized independently using default parameters via Seurat::NormalizeData. The log-normalized expression values are then z-scored on the integrated object via Seurat::ScaleData using default parameters after finding anchoring cells between samples.

## Dimensional reduction, and cell classification

Choosing an optimal number of principal components (PCs) for dimensional reduction was determined by scree plot using Seurat::ElbowPlot. We selected PCs showing the greatest variance explained until each subsequent PC showed little to no change. For the macrophage time course, we specified 50 total number of PCs to compute and selected the first 49 PCs based on the scree plot to build a shared nearest neighbor (SNN) graph using Seurat::RunPCA and Seurat::FindNeighbors, respectively. Seurat::FindClusters was used with a resolution of 0.8, resulting in 31 clusters. To visualize cells in two-dimensional latent space, we used UMAP dimensional reduction technique via Seurat::RunUMAP using the first 49 PCs aforementioned.

For the *il10ra* dataset, we specified 50 total number of PCs to compute and selected the first 28 PCs based on the scree plot to build a shared nearest neighbor (SNN) graph using Seurat::RunPCA and Seurat::FindNeighbors, respectively. Seurat::FindClusters was used with a resolution of 0.8, resulting in 30 clusters. To visualize cells in two-dimensional latent space, we used UMAP dimensional reduction technique via Seurat::RunUMAP using the first 28 PCs aforementioned.

Classification of cell clusters for the macrophage time course and *il10ra* sibling versus mutant dataset were annotated by calculating differential marker expression via Seurat::FindAllMarkers using default parameters.

## DE analysis or primary analysis

To distinguish differentially expressed cluster marker genes for both the macrophage time course and the *il10ra* dataset, we used Seurat::FindMarkers to compare cells in one query time point against all other cells (Supplementary Data 1 and 3). For all differentially expressed gene tables, we defined our statistical test using Wilcoxon Rank-test. Only genes with a p-value less than 0.05 were retained. Venn diagrams for Supplementary Data 2 were generated using Manteia[87].

## HCR-FISH

Hybridization chain reaction (HCR) was performed according to manufacturer's instructions for *il10ra*-B2 (4pmol), *fgl2a*-B2 (2pmol), *irg1*-B2 (2pmol), *mcamb*-B2 (2pmol), *f13a1b*-B2 (2pmol), *tspan10*-B2 (2pmol), *runx3*-B2 (2pmol), *eomesa*-B2 (2pmol), *hepacam2*-B2 (2pmol), *dusp1*-B2 (2pmol) and mScarletI-B4 (2pmol)[39] (Molecular Instruments) except that probes were incubated for 48 h and amplifiers incubated for 48 h. The amplifiers used were B2-546 and B2-647 (Molecular

Instruments). HCR fish were subsequently stained with DAPI (5ug/mL) for 30 min at room temperature (RT) in the dark and washed three times with 5x SSCT before imaging. Quantification was made by visually assessing colocalization between the GFP signal from the *mpeg:GFP* macrophages and the saturated HCR signal using FIJI[88] manually in the 3D z-Stack. No threshold parameters were used, and HCR signal was scored as either present or absent within the GFP signal in 3D.

### Time-lapse and confocal imaging
Images were acquired using a Nikon Ti Eclipse with Yokogawa CSU-W1 spinning disk head equipped with a Hamamatsu Flash 4.0 sCMOS. Objective lenses used were a Nikon Plan Apo 40 × 1.15 NA LWD (water) and a Nikon Plan Apo 20 × 0.75 NA.

For live imaging experiments, larvae were immobilized with tricaine (MS-222) up to 150 mg/L and mounted in glass bottom dishes (MatTek) with 0.8% low melting point agarose dissolved in 0.5x E2 with tricaine (100 mg/L). Time lapse recordings were started 10-min after addition of neomycin (300μM) on top of the agarose. Temperature was kept constant at 28.5 °C using a Stage Top Chamber (OkoLab).

A Nikon LUNV solid state laser launch was used for lasers 405, 445, 488, 561, and 647 nm. Emission filters used on the Nikon were 480/30, 535/30, 605/70.

All image acquisition was performed using Nikon Elements AR 4.6 (Nikon) software.

### Macrophage recruitment assay
5dpf larvae expressing *mpeg:GFP* or *irg1:lck-mScarletI* or *stat1b:lck-mScarletI* and *she:lckmScarletI* or *she:GAP43-GFP* were treated with 300μM of neomycin for 30 min and fixed in 4% PFA after 1 h, 3 h, 5 h and 7 h after. Larvae were subsequently imaged using spinning disk confocal microscopy. Quantifications of macrophages inside the neuromast was performed in 3D in Imaris 9.0. Each experiment has been performed 3 times independently.

### *il10ra/stat6* double mutant assays
Double heterozygous larvae for *il10ra* and *stat6* were incrossed to analyze WT, single homozygous *il10ra*, single homozygous *stat6* and double homozygous *il10ra/stat6* larvae.

Macrophage recruitment assay: 48 larvae at 5 dpf were analyzed for each condition (homeo, neo+1 h, neo+3 h, neo+5 h) were fixed in 4% PFA O/N at 4 C, washed 3 times in PBS. The head was used for PCR genotyping while the trunk was kept for imaging. Larvae were genotyped by PCR + electrophoresis for *il10ra* using a mix of 3 primers to identify homozygous vs heterozygous or WT using the following primers: Fwd deletion: 5-TGCATAACAGCTCAGCCATCTTCTC-3; Rv: 5-CATGTCACATTCTTCCACAATGTTCC-3; Fwd-hets: 5-GAATCGACGGAGCTAGTAAGAGC-3. Homozygosity results in a 250 bp band, heterozygosity in a 327 bp and a 250 bp band. WT in a 327 bp band. Larvae were genotyped by PCR + sanger sequencing to identify the 10 bp deletion using the following primers: Fwd 5-ACAACTTGTGAGGACAGTTCGG-3; Rev 5- GGAACTGAATTGGGAGAAAGAGACG-3 resulting in a 360 bp fragment.

HC/synapse regeneration assay: 48 larvae at 5 dpf were analyzed for each condition (homeo, neo+24 h, neo+48 h) were fixed in 4%PFA O/N at 4 C and washed 3 times in PBS. The head was used for PCR genotyping, as described above, while the trunk was kept for immunofluorescence. The efferent synapses were labeled with Vamp2 and tested for interactions with a HC labeled with Otoferlin using Imaris version 9.6 in 3D. All genotyping were performed using GoTaq (Promega #M512B; following manufacturer protocol) using the following program: 95 C for 5 min then 30 cycles of 95 C for 30 s, 59 C for 30 s and 72 C for 1 min followed by a final extension step at 72 C for 2 min.

### Whole-mount immunofluorescence
Larvae were fixed in 4%PFA O/N at 4 C, washed 3 times in PBS, blocked for 2 h in 5% GS/PBSTritonX1%, incubated O/N at RT in primary antibody solution (2%GS/PBSTritonX1%). The following day, larvae were washed 4 times for 1 h in PBSTween0.1%, incubated in secondary antibody solution (PBSTritonX1%) for either 2 h at RT or O/N at 4 C. Primary antibodies: HCS-1 (otoferlin from Hybridoma Banks) 1/500 dilution; VAMP2 (Genetex, #GTX132130) 1/500 dilution. Secondary antibodies: anti-mouse_Alexa546 1/1000; anti-rabbit_Alexa488 1/1000 and DAPI 1/2000.

### Macrophage photoconversion
A circular region of interest was drawn on top of the neuromast 1H after neomycin to only photoconvert effector macrophages. Photoconversion was performed using the FRAP module of the Nikon spinning disk confocal. We used a 405 laser at 3% and a dwell time of 300 μs. Stimulation was repeated twice to achieve a good photoconversion. The photoconversion is expected to occur in 30–40% of the protein. The rest will be photoactivated (increased green fluorescence) and will eventually bleach. Then each photoconverted neuromast was imaged with the 445, 488, and 561 nm lasers with a z-stack constant of 50 μm from the middle of the neuromast and a zStep of 1 μm (51 slices) with a 40xLWD 3 h and 5 h after neomycin. Quantification was performed in Imaris 9.0.

### Macrophage distance and velocity quantifications
Distances and velocities were extracted from time-lapse recordings of *mpeg:GFP* and *she:lck-mScarletI* 5dpf larvae. The center of the neuromast as well as individual macrophages were tracked over time using Imaris 9.0. Both values were exported form the statistics module of Imaris. Graphs were made using GraphPad Prism 8 (version 8.4.3).

### Drug treatment for GR inhibition
5dpf *mpeg:GFP* larvae were pre-treated with 10 μM RU486 (Sigma M8046) in 0.01%EtOH/0.5xE2 or 0.01%EtOH/0.5xE2 (control) for four hours prior to neomycin treatment. Larvae were kept in RU486 during and after neomycin treatment until fixation in 4%PFA at 4 C.

### GO and pathway analysis
Gene ontology (GO) and pathway enrichment analysis was performed using Metascape[42].

### Statistical analysis
All statistical tests were performed using GraphPad Prism 8 (version 8.4.3) as indicated in the figure legends. When comparing data from more than two groups, statistical significance was calculated using one-way ANOVA with Tukey's post hoc test. Data from two groups were compared using two-tailed unpaired t-test. *p-value*s smaller than 0.05 were considered to be statistically significant. Plots were made in GraphPad Prism 8.

### Reporting summary
Further information on research design is available in the Nature Research Reporting Summary linked to this article.

## Data availability
Original data underlying this manuscript can be accessed from the Stowers Original Data Repository at http://www.stowers.org/research/publications/libpb-1663. In addition, Source data for each graph is provided with this paper as a Source Data File. Raw and processed scRNAseq data generated in this study have been deposited in the Gene Expression Omnibus database (GEO #GSE209884) Source data are provided with this paper.

## Code availability

Code for the SeuratExtensions used to process/analyze scRNAseq data can be found: https://doi.org/10.5281/zenodo.6871552.

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

## Acknowledgements

We thank the Piotrowski lab members for insightful discussions and Dr. Ruslan Medzhitov, Dr. Mark Lush, Julia Peloggia, and Daniela Münch for critical reading of the manuscript. We thank Allison Scott for technical assistance. We are also grateful to the Stowers Institute Core Facilities (Aquatics team, Cytometry Core, Imaging Core, Molecular Biology Core) for their technical expertise. We thank Dr. Nels Elde and Keir Balla for the kind gift of the pCS2-InfPhi1 plasmid. This work was funded by an NIH (NIDCD) award 1R01DC015488-01A1 and by institutional support from the Stowers Institute for Medical Research to T.P.

## Author contributions

N.D. designed and performed the experiments, analyzed, and interpreted the data, and wrote the manuscript; N.T. analyzed and interpreted the scRNAseq data; D.D. analyzed and interpreted the scRNAseq data; M.S. performed experiments; J.B. performed experiments; T.P. designed the experiments, analyzed and interpreted the data, and wrote the manuscript.

## Competing interests

The authors declare no competing interests.
