## [Peer Review File · Nature Communications]

An anti-inflammatory activation sequence governs
macrophage transcriptional dynamics during tissue injury in
zebrafishREVIEWER COMMENTS

Reviewer #1 (Remarks to the Author):

This interesting manuscript contains an important temporal analysis of macrophage phenotype during lateral line regeneration zebrafish larvae. The findings are potentially of significance, but there are some areas where additional discussion and citation would improve the manuscript.

The authors have initially characterised macrophage recruitment to neomycin damaged hair cells, then performed scRNAseq on sorted mpeg transgenic positive cells from these larvae. The resulting data and analysis has considerable value, particularly as it shows three distinct antiinflammatory phenotypes. It is somewhat surprising that these are not interdependent, but the authors have tested this with pharmacological or genetic means and shown that the three populations can still occur. Some of the findings are validated using fluorescent in situ hybridisation to clarify the cell type involved - an elegant experiment.

My concerns are, that this is a very specific example of regeneration, which might not be generalisable to other models. Can the authors interrogate any other datasets to confirm that their findings are generalisable to other systems? This limitation should be discussed.

The first section of the results is titled to suggest that inflammation resolves over a 5 hour time course. However, other than macrophage numbers, I did not see any measures of inflammation. These should either be included, or the text amended accordingly.

Finally, are there clues in the dataset which might help understand why any particular macrophage takes on a particular molecular phenotype? It would be interesting to know whether the authors can comment on this.

Reviewer #2 (Remarks to the Author):

This manuscript by Piotrowski and colleagues, "An anti-inflammatory activation sequence governs macrophage transcriptional dynamics during tissue injury," describes the transcriptional dynamics of macrophages responding to chemically induced injury of hair cells of the lateral line of larval zebrafish. The investigators take advantage of the visual accessibility and repeated structures of the hair cell containing neuromasts to obtain a high temporal resolution description of the macrophage dynamics, coupled with single cell transcriptomics, revealing three temporally distinct patterns of anti-inflammatory gene expression of these macrophages. They then go on to demonstrate that the latter two gene expression patterns do not require the early programs. Specific comments follow.

1. This manuscript describes a technical a tour de force of description of macrophages during tissue repair at high temporal and spatial resolution. The authors use specific pharmacological and genetic reagents to demonstrate the lack of requirement of the earlier programs of glucocorticoid and interleukin-10 signaling for the latter gene expression programs. A disappointment of the study is that the authors fail to comment on whether any of the anti-inflammatory macrophage programs they describe are required for tissue repair. Given that they have the reagents in hand to inhibit glucocorticoid and interleukin-10 signaling, it is puzzling that they did not test whether animals in which these programs are inhibited have any defects in neuromast regeneration or earlier steps such as cell debris clearance.

2. The authors take advantage of the fact that larval zebrafish have multiple neuromasts that can be analyzed, increasing the power of their analysis. However, they treat the individual neuromasts from the same fish as independent measurements equivalent to independent neuromasts from different fish (e.g. graphs in Fig 1 C,D and Fig2 E,I,M). It may be the case that the neuromasts from an individual fish are more likely to have similar properties to each other than the neuromasts from different fish. The authors should perform a multivariate analysis of their data to establish whether this

assumption is accurate.

3. The authors use HCR-FISH to validate their single cell transcriptomics of macrophages. They report their analysis (for example in Fig 2 F,J,N) as a ratio of the GFP positive macrophages with HCR signal, but we are not provided with an expectation to interpret these ratio. For example, if they performed HCR with a mpeg probe, would they observe a ratio of exactly 1, or is the HCR less efficient than that? Related to this, the authors should provide more information on how they performed quantification of their HCR-FISH rather than the single line on 455.

Reviewer #3 (Remarks to the Author):

In this manuscript, "An anti-inflammatory activation sequence governs macrophage transcriptional dynamics during tissue injury", Denans et al investigate macrophage responses to hair cell injury, using both live timelapse imaging and scRNAseq. They find that a small number of macrophages are recruited to injured sites, and that these initially recruited cells persist and directly interact with the injured hair cells for at least 5 hours. Additionally, scRNAseq shows progressive transcriptional activation programs in the macrophages. Overall this is a very well-written and well-presented study with a strong premise. How macrophages promote tissue repair/regeneration is largely unknown and this study sheds insight on this question. However, I have some concerns over a lack of experimental details and replicates. Additionally, the importance of these transcriptional programs in tissue repair is not explored at all and the relevance of macrophage responses to the systemic damage caused to all hair cells in the larvae versus a local injury is unknown.

Major issues

1. There is lots of scRNAseq data presented, but the authors manage to narrow down the focus to 3 core transcriptional programs. Additionally, two of these programs are inhibited (by drug treatment or genetic mutation) to demonstrate that these signaling programs are independently activated. However, the key question is whether these signaling pathways affect wound repair/regeneration. The tools are already in hand to perform these experiments and neuromast/hair cell regeneration should be quantified when these signaling networks are inhibited. The last line of the discussion is that "This finding has important implications for the design of targeted immunomodulatory therapies." But without this data of whether these pathways actually influence the outcome of regeneration there are few implications.

2. The hair cell ablation method used or antibiotic treatment is reliable and consistent and causes multiple injuries throughout the fish, making the scRNAseq experiment from sorted macrophages from whole larvae possible. However, the results may be affected by this type of systemic injury versus a more localized injury at one neuromast- while such a localized injury is more likely to mimic a real human situation. For example, maybe no second wave of macrophages is observed because all macrophages that can respond have already been recruited to a different hair cell injury. A targeted laser injury should be done and live imaged to confirm the macrophage behavior results. Concomitant HCT to confirm similar transcriptional programs with a targeted injury would also strengthen the paper.

3. Experimental replicates: In figure legends state the number of larvae that were analyzed per condition, but it is unclear if these were done all on the same day or as multiple replicates. Variation in larvae can happen day to day and clutch to clutch, so all experiments should be done in at least 3 independent replicates. (All Fig 1, Fig 2 E-O, 3 D,F, 4 A-E).

4. Almost no details are provided on the generation of new zebrafish lines.

-New transgenic lines: *irg1* and *stat1b* reporters. Cloning details and primer sequences for promoters must be added. Information on what was injected into embryos to establish these lines and how they were screened must also be added. A control is also needed that demonstrates these lines truly report the expression of the targeted genes. In general, the methods section on fish lines would benefit from a table of all lines, their purpose, and a reference for the first publication (this information

is also needed for all lines) would be clearer.

-il10ra CRISPR mutant: The sgRNA sequences are written out, but how were these sgRNAs produced in the lab? How much was injected into embryos? With what Cas9? What fish background was used? How were mutations identified and screened? What is the actual sequence of the mutated site in this new line? Fig 4G which shows a lack of HCR staining requires a positive control. How were homozygous mutant and heterozygous larvae generated for experiments- what crosses were done?

Minor issues

1. Some of the set up of the premise of the study in the introduction is a little unclear. I think the idea is to define the transcriptional program(s) of macrophages after injury in a regenerating organism (larval zebrafish) and that then activating these programs in non-regenerating organisms might promote regeneration over scarring. However, this is not clearly laid out in the introduction. There is some conflation of regenerating and non-regenerating animals and what different macrophage programs might be doing in each of these contexts.

2. The authors claim that their scRNAseq analysis shows that macrophages go through a “linear sequence” (line 213) of activation states and that “the same macrophages transition through different anti-inflammatory states” (line 218). I don’t see the data for this. Yes, as a population the cells are clearly making these transitions but there is no direct evidence to know for sure it is the same cells. Would be more convinced if it was reported from the scRNAseq data what percentage of macrophages express each marker of interest. Is it actually that close to 100% of macrophages express each? Or is it less than 50%? I believe this data should be available in the scRNAseq dataset but it doesn’t appear to be presented in the paper.

3. In the introduction, talk about the limitations of the M1 vs M2 paradigm. I totally agree with the authors that this paradigm is an oversimplification but think some additional information would strengthen this argument and applies directly to this study system. One issue with this paradigm in zebrafish larvae is that these phenotypes were partially defined by the polarization of T cells that activate the macrophages. (Th1 - M1 and Th2 - M2). But larvae don’t have T cells, making this type of T cell-mediated in vivo activation impossible. Additionally, in zebrafish studies (and other animal models and cell culture studies) “M1” macrophages often just means Tnfa+ or some other marker gene. As the authors argue, there could be multiple networks that are different that each activate such a marker gene.

4. I am confused about the naming and quantification of “effector” and “non-effector” macrophages (line 97). How are these two classes defined? Isn’t the definition whether they migrate to the neuromast or not? Then why is it interesting that the effectors are closer in Fig 1E--isn’t this just the definition?

5. In many graphs the data points are so large that it is impossible to see them individually. Should be make smaller so that it doesn’t just appear as clumps of dots (Fig 1 C,D 2 E-N, 3 D,F).

6. What are the cells Fig 1H that are dendra+ but don’t get photoconverted?

7. In the scRNAseq, find cells that are not macrophages (line 121). It is possible that the FACS was just not specific and some non-GFP+ cells got through and were sequenced. FACS gating plots are not shown. Does the scRNAseq data actually show that these other cell types express GFP and mpeg1?

8. State in discussion (line 260) the pro-inflammatory signals “attract more macrophages to the injury site if the organ does not possess a resident population, or if the size of the injury requires more macrophages” . This isn’t really correct. First, basically all organs have a resident macrophage population. Second, in fully developed organisms, monocytes must be recruited from the blood and these monocytes differentiate into macrophages in the tissue.

9. State in discussion (line 275) that GR activation leads to chromatin unwinding and suggest that this “likely turns off the transcription of pro-inflammatory cytokines”. I don’t understand this- isn’t a permissive chromatin environment associated with increased transcription, not decreased?. Isn’t it more likely that the GR activation is inhibiting NF- κ B as many of the “pro-inflammatory” genes (ex IL1b) are activated by NF- κ B?

10. Line 287: Describe results after “the loss of IL10 in zebrafish”. Should clarify there is a loss of signaling via the IL10ra receptor, not the loss of the cytokine itself in these experiments.

Point by point response to the reviewers

We appreciate the time and effort the reviewers took in crafting their thoughtful comments and for their overall very positive evaluation of our manuscript. We have taken the reviewers' comments and suggestions to heart and performed additional experiments, added figures and clarified or rephrased the text (highlighted in yellow). Thanks to the reviewers' comments the manuscript is now significantly improved.

Reviewer #1 (Remarks to the Author):

This interesting manuscript contains an important temporal analysis of macrophage phenotype during lateral line regeneration zebrafish larvae. The findings are potentially of significance, but there are some areas where additional discussion and citation would improve the manuscript.

The authors have initially characterised macrophage recruitment to neomycin damaged hair cells, then performed scRNAseq on sorted mpeg transgenic positive cells from these larvae. The resulting data and analysis has considerable value, particularly as it shows three distinct antiinflammatory phenotypes. It is somewhat surprising that these are not interdependent, but the authors have tested this with pharmacological or genetic means and shown that the three populations can still occur. Some of the findings are validated using fluorescent in situ hybridisation to clarify the cell type involved - an elegant experiment.

My concerns are, that this is a very specific example of regeneration, which might not be generalisable to other models. Can the authors interrogate any other datasets to confirm that their findings are generalisable to other systems? This limitation should be discussed.

Response: We agree that it would be very interesting to determine if the same sequence of anti-inflammatory signaling occurs in other regenerating systems. Unfortunately, to our knowledge, no such closely timed scRNASeq analyses of macrophage responses in other regenerating organs yet exist. There are hints that this sequence could be involved in the resolution of injury in other injury types and species. For instance, it has recently been shown that an acute heart injury in the adult zebrafish triggers a glucocorticoid pathway activation in undamaged distant organs (Sun et al., Nat Cell Bio 2022). Moreover, both IL10 and IL4 signaling have been shown to be activated in response to muscle injury in mouse macrophages (Deng et al., J Immunology, 2012). Interestingly, a very recent study comparing the transcriptome of regenerating African spiny mouse (*Acomys*) versus non regenerating *Mus Musculus* in a model of spinal cord injury highlights the activation of IL10 and IL4 signaling during regeneration (Nogueira-Rodrigues et al, Dev Cell, 2022). Unfortunately, these studies lack both the temporal and spatial resolution necessary to draw a conclusion. We have now added a paragraph in the discussion section in which we highlight both the potential conservation of the macrophage anti-inflammatory sequence, as well as the limitations of our study.

The first section of the results is titled to suggest that inflammation resolves over a 5 hour time course. However, other than macrophage numbers, I did not see any measures of inflammation. These should either be included, or the text amended accordingly.

Response: There is, unfortunately, no good consensus definition of inflammation. We described the resolution of inflammation corresponding to immune cells leaving the damaged organ. However, we agree that the term inflammation in this case can be confusing and thus we rephrased the section title to: "the same population of effector macrophages invades and leaves the neuromast within a five-hour window".

Finally, are there clues in the dataset which might help understand why any particular macrophage takes on a particular molecular phenotype? It would be interesting to know whether the authors can comment on this.

Response: We did not find any molecular clue in our dataset, such as specific *tlr* receptors for instance, that would point toward a specific population of macrophage being the only one able to become the effector population. Our spatial analysis shows that the closest macrophages to the neuromast become effectors. We therefore hypothesize that all macrophages are plastic and the first ones that receive signals from dying cells are activated.

Reviewer #2 (Remarks to the Author):

This manuscript by Piotrowski and colleagues, "An anti-inflammatory activation sequence governs macrophage transcriptional dynamics during tissue injury," describes the transcriptional dynamics of macrophages responding to chemically induced injury of hair cells of the lateral line of larval zebrafish. The investigators take advantage of the visual accessibility and repeated structures of the hair cell containing neuromasts to obtain a high temporal resolution description of the macrophage dynamics, coupled with single cell transcriptomics, revealing three temporally distinct patterns of anti-inflammatory gene expression of these macrophages. They then go on to demonstrate that the latter two gene expression patterns do not require the early programs. Specific comments follow.

1. This manuscript describes a technical a tour de force of description of macrophages during tissue repair at high temporal and spatial resolution. The authors use specific pharmacological and genetic reagents to demonstrate the lack of requirement of the earlier programs of glucocorticoid and interleukin-10 signaling for the latter gene expression programs. A disappointment of the study is that the authors fail to comment on whether any of the anti-inflammatory macrophage programs they describe are required for tissue repair. Given that they have the reagents in hand to inhibit glucocorticoid and interleukin-10 signaling, it is puzzling that they did not test whether

animals in which these programs are inhibited have any defects in neuromast regeneration or earlier steps such as cell debris clearance.

Response: While the premise of our study was to unveil the molecular sequence of macrophage activation, we agree with the reviewer that testing the impact of such activation on tissue regeneration will greatly add to our study. Since the GR activation is systemic, and manipulation of GR signaling as been shown to directly affect HC regeneration (Namdaran et al., 2012 Journal of Neuroscience), we focused on manipulating IL10 and IL4 signaling, which are more leukocyte specific in our context. We have now analyzed hair cell regeneration in *il10ra* and *stat6* single mutants, which both have normal hair cell regeneration. To test if IL10 and IL4 signaling might act redundantly we generated double mutants to inhibit both IL10 (*il10ra* KO) and IL4 (*stat6* KO) signaling. A hallmark of sensory hair cell differentiation is their capacity to form synapses with the lateral line nerve. We thus quantified the number of presynaptic buttons using an antibody against VAMP2, as well as the number of hair cells 24h and 48h after hair cell death, during the regeneration process. This analysis shows that while the number of HC does not significantly differ in neither single or double (*il10ra/stat6*) mutants, the number of presynaptic buttons is dramatically reduced (by 50%) only in the double mutants suggesting a synergistic role for IL10 and IL4 signaling in hair cell synapses formation during regeneration. We also quantified the recruitment of macrophages in the double mutant and found it unaffected suggesting that these pathways do not control macrophage dynamics after injury but the regeneration process. This new data has been added in the form of Figure 5 and Supplementary Figure 10.

2. The authors take advantage of the fact that larval zebrafish have multiple neuromasts that can be analyzed, increasing the power of their analysis. However, they treat the individual neuromasts from the same fish as independent measurements equivalent to independent neuromasts from different fish (e.g. graphs in Fig 1 C,D and Fig2 E,I,M). It may be the case that the neuromasts from an individual fish are more likely to have similar properties to each other than the neuromasts from different fish. The authors should perform a multivariate analysis of their data to establish whether this assumption is accurate.

Response: We always use the exact same number of neuromasts per fish, thus our statistical analysis takes into account the weight of the variability between fish. To test the reviewer's hypothesis, we have now analyzed the variability between fish and do not find a statistical difference compared to our previous analysis. Please find an example below where we plotted the data from Fig1d either from individual neuromasts or per fish.

Figure R1. Comparison of variability between neuromasts vs larvae

3. The authors use HCR-FISH to validate their single cell transcriptomics of macrophages. They report their analysis (for example in Fig 2 F,J,N) as a ratio of the GFP positive macrophages with HCR signal, but we are not provided with an expectation to interpret these ratio. For example, if they performed HCR with a mpeg probe, would they observe a ratio of exactly 1, or is the HCR less efficient than that? Related to this, the authors should provide more information on how they performed quantification of their HCR-FISH rather than the single line on 455.

Response: We apologize for not explaining our analysis properly. The rationale for using HCR in Fig2 was to identify the clusters of macrophages that invade the neuromast and phagocytose dead HC (effector macrophages), not to provide an absolute value for the involvement of each population. As any other ISH methods, the intensity and quality of each HCR depends on the quality of the probe set (same is true for q-PCR), thus we cannot provide an expectation for the sensitivity of each probe. Most of these genes are expressed in other tissues and we consider the probe “working” if we can detect the described expression. Our HCR protocol is aimed at saturating the signal (we perform probe development for 2 days as described in the method section). Thus, we do not quantify single molecule but perform a binary quantification (present vs absent). Regarding the quantification since we are analyzing 3D images, we visually quantify the presence or absence of signal by going through each slice of the z-stack in FIJI using the GFP signal to delineate the macrophage. We have now added more information of the process in the method section)

Reviewer #3 (Remarks to the Author):

In this manuscript, “An anti-inflammatory activation sequence governs macrophage

transcriptional dynamics during tissue injury”, Denans et al investigate macrophage responses to hair cell injury, using both live timelapse imaging and scRNAseq. They find that a small number of macrophages are recruited to injured sites, and that these initially recruited cells persist and directly interact with the injured hair cells for at least 5 hours. Additionally, scRNAseq shows progressive transcriptional activation programs in the macrophages. Overall this is a very well-written and well-presented study with a strong premise. How macrophages promote tissue repair/regeneration is largely unknown and this study sheds insight on this question. However, I have some concerns over a lack of experimental details and replicates. Additionally, the importance of these transcriptional programs in tissue repair is not explored at all and the relevance of macrophage responses to the systemic damage caused to all hair cells in the larvae versus a local injury is unknown.

Major issues

1. There is lots of scRNAseq data presented, but the authors manage to narrow down the focus to 3 core transcriptional programs. Additionally, two of these programs are inhibited (by drug treatment or genetic mutation) to demonstrate that these signaling programs are independently activated. However, the key question is whether these signaling pathways affect wound repair/regeneration. The tools are already in hand to perform these experiments and neuromast/hair cell regeneration should be quantified when these signaling networks are inhibited. The last line of the discussion is that “This finding has important implications for the design of targeted immunomodulatory therapies.” But without this data of whether these pathways actually influence the outcome of regeneration there are few implications.

Response: We agree with reviewers 2 and 3 that testing the impact of the anti-inflammatory pathways that were uncovered in our scRNAseq on tissue regeneration will greatly add to our study. Since the GR activation is systemic, and manipulation of GR signaling as been shown to directly affect HC regeneration (Namdaran et al., 2012 Journal of Neuroscience), we focused on manipulating IL10 and IL4 signaling, which are more leukocyte specific in our context. We have now analyzed hair cell regeneration and macrophage dynamics in *il10ra* and *stat6* single and double mutants. Surprisingly, both HC number and macrophage dynamics are unaffected in single and double mutants. Since macrophages have been shown to be required for synaptogenesis after HC death in the mouse cochlea, we quantified the number of presynaptic buttons, using an antibody against VAMP2, at 5dpf, 24h and 48h after hair cell death, during the regeneration process. This analysis shows that the number of presynaptic buttons is dramatically reduced (by 50%) only in the double mutants suggesting a synergistic role for IL10 and IL4 signaling in hair cell synapses formation during regeneration but not during development. This new data that provide a new role for these established anti-inflammatory pathways, has been added in the form of Figure 5 and Supplementary Figure 10.

2. The hair cell ablation method used or antibiotic treatment is reliable and consistent and causes multiple injuries throughout the fish, making the scRNAseq experiment from sorted macrophages from whole larvae possible. However, the results may be affected by this type of systemic injury

versus a more localized injury at one neuromast- while such a localized injury is more likely to mimic a real human situation. For example, maybe no second wave of macrophages is observed because all macrophages that can respond have already been recruited to a different hair cell injury. A targeted laser injury should be done and live imaged to confirm the macrophage behavior results. Concomitant HCT to confirm similar transcriptional programs with a targeted injury would also strengthen the paper.

Response: We agree with the reviewer that it is important to contrast systemic injuries with localized injuries and we now discuss the possible differences between a localized and a systemic injury in the Discussion section on page 12. However, we do not perceive the fact that we are inducing a systemic injury as a limitation of our study. Systemic chemical injuries also occur in humans as the result of antibiotic treatment and chemotherapy that are commonly used to treat human diseases such as severe bacterial infections and cancer.

Irrespective, we agree that it would be interesting to compare a systemic with a local injury, however performing these experiments in a rigorous fashion is beyond the scope of this manuscript. The reason is that laser ablation will trigger a different type of cell death leading to different Damage Associated Molecular Patterns that can lead to a different macrophage activation than neomycin induced cell death. The detection of differences in the activation states would require a quantitative in depth studies requiring scRNA-seq experiments that are technically impossible using the 10X technology because of the low number of cells that can be collected after laser ablation. In addition, while the major pathways might be conserved, the downstream targets could be different and drawing conclusions based on a few markers via HCR would be misleading.

Furthermore, HCs are not located in the same z-plan, thus laser ablation would require to ablate on average 18 to 20 HCs one by one using a multiphoton laser without damaging the support cells (SCs) whose apical extensions reside between each HCs. This is extremely technically challenging especially without fluorescent markers that can discriminate HCs and SCs. Damaging SCs will have a profound effect on the regenerative process since they are the progenitors cells that will produce new HCs.

Lastly, we do not believe that the lack of a second wave of macrophages is due to their depletion because all neuromasts are affected leading to an exhaustion of macrophages capable of responding to HC death. Fig1C and 1F show recruitment of macrophages toward the neuromast that will not participate in HC clearance suggesting that these cells can sense the dying HC signal but do not invade the neuromast. Thus, even in the presence of other macrophages surrounding the neuromasts, we never observe more than 3 to 4 effector macrophages inside the neuromasts. This might be a mechanism to prevent more damage to the neuromast by limiting the number of macrophages involved in the repair/regeneration process.

3. Experimental replicates: In figure legends state the number of larvae that were analyzed per condition, but it is unclear if these were done all on the same day or as multiple replicates. Variation in larvae can happen day to day and clutch to clutch, so all experiments should be done in at least 3 independent replicates. (All Fig 1, Fig 2 E-O, 3 D,F, 4 A-E).

Response: We have now clarified the experimental design in the figure legend. For each experiment we collect several clutches that are pooled to prevent clutch-based biases. All experiments have been performed as biological triplicates (different days) except for the scRNAseq (1 dataset per timepoint) and Fig5 (4 biological replicates). We also have extended our description of the experimental designs in the Method section.

4. Almost no details are provided on the generation of new zebrafish lines.

-New transgenic lines: *irg1* and *stat1b* reporters. Cloning details and primer sequences for promoters must be added. Information on what was injected into embryos to establish these lines and how they were screened must also be added. A control is also needed that demonstrates these lines truly report the expression of the targeted genes. In general, the methods section on fish lines would benefit from a table of all lines, their purpose, and a reference for the first publication (this information is also needed for all lines) would be clearer.

-*il10ra* CRISPR mutant: The sgRNA sequences are written out, but how were these sgRNAs produced in the lab? How much was injected into embryos? With what Cas9? What fish background was used? How were mutations identified and screened? What is the actual sequence of the mutated site in this new line?

Response: We apologize for the brevity of the method section and thank the reviewer for the detailed request which makes our method section more reproducible. We have now added more detailed information in the method section on page 13. We also included a table for all the transgenic lines generated as suggested.

We also demonstrate that the *irg1* reporter line responds to LPS injection and our *stat1b* reporter is a faithful reporter of Interferon activation (see Supplementary Fig4).

For the *il10ra* mutant, we deleted the promoter to completely abolish transcription as validated by HCR (See Supplementary Fig7) and in the scRNAseq experiment. We have also added all the requested details about the sgRNA (synthetic from IDT), the genotyping procedure, the fish background and the injection parameters.

We also added the feeding regimen of our adult fish since this can affect the type of challenges the macrophages face and thus their activation status. For instance, live feed, such as artemia, may contain pathogens and result in a pathogen response state in macrophages unlike dry feeds.

Fig 4G which shows a lack of HCR staining requires a positive control. How were homozygous mutant and heterozygous larvae generated for experiments- what crosses were done?

Response: We thank the reviewer for catching our mistake regarding the positive control of former Fig 4G. It has been removed during figure prepping due to lack of space. We have now moved this result in Supplementary Fig7 with the appropriate positive control. We apologize for the lack of experimental details and have added more detailed information in the method section on page 13. Homozygous larvae were generated by an incross of homozygous fish and the heterozygous larvae by an outcross of homozygous fish with WT (AB strain). We used this strategy since homozygous larvae do not show any visual phenotype and this is the same experimental plan as for the scRNA-seq of the *il10ra* mutant.

Minor issues

1. Some of the set up of the premise of the study in the introduction is a little unclear. I think the idea is to define the transcriptional program(s) of macrophages after injury in a regenerating organism (larval zebrafish) and that then activating these programs in non-regenerating organisms might promote regeneration over scarring. However, this is not clearly laid out in the introduction. There is some conflation of regenerating and non-regenerating animals and what different macrophage programs might be doing in each of these contexts.

Response: We simplified the introduction based on the thoughtful recommendations of the reviewer by removing the back and forth between regenerating and non-regenerating species and streamlining the premise of the study.

2. The authors claim that their scRNAseq analysis shows that macrophages go through a “linear sequence” (line 213) of activation states and that “the same macrophages transition through different anti-inflammatory states” (line 218). I don’t see the data for this. Yes, as a population the cells are clearly making these transitions but there is no direct evidence to know for sure it is the same cells. Would be more convinced if it was reported from the scRNAseq data what percentage of macrophages express each marker of interest. Is it actually that close to 100% of macrophages express each? Or is it less than 50%? I believe this data should be available in the scRNAseq dataset but it doesn’t appear to be presented in the paper.

Response: We agree with the reviewer that we do not have evidence of the macrophage transition states at the single cell level but only at the population level. We have now amended our statement accordingly in the text to highlight this limitation. Due to the low sequencing depth and high dropout rate of any scRNAseq experiments, percentage of cells expressing a gene is not a faithful quantitative readout. The DotPlot in Fig3 provides the percentage of effector cells expressing each gene in the effector macrophages population. For *il1ora* and *fgl2a* for which we have HCR, this percentage is much lower in the scRNAseq (30%) than in the HCR quantifications (80%) (more sensitive method) (see Fig3).

3. In the introduction, talk about the limitations of the M₁ vs M₂ paradigm. I totally agree with the authors that this paradigm is an oversimplification but think some additional information would strengthen this argument and applies directly to this study system. One issue with this paradigm in zebrafish larvae is that these phenotypes were partially defined by the polarization of T cells that activate the macrophages. (Th₁ - M₁ and Th₂ - M₂). But larvae don’t have T cells, making this type of T cell-mediated in vivo activation impossible. Additionally, in zebrafish studies (and other animal models and cell culture studies) “M₁” macrophages often just means Tnfa⁺ or some other marker gene. As the authors argue, there could be multiple networks that are different that each activate such a marker gene.

Response: We thank the reviewer for this comment and amended this part of the introduction based on their suggestion.

4. I am confused about the naming and quantification of “effector” and “non-effector” macrophages (line 97). How are these two classes defined? Isn’t the definition whether they migrate to the neuromast or not? Then why is it interesting that the effectors are closer in Fig 1E-- isn’t this just the definition?

Response: We apologize for not being clear in our definition. The effector population is defined as the population that invade the neuromast and phagocytose dead HCs. Some non-effector macrophages are recruited toward the neuromast but never invade it or participate in phagocytosis. We have amended the text accordingly. We believe that the fact that effector macrophages are localized closer to the neuromast prior to HC death is interesting because it might suggest that there is no particular molecular state that makes an effector competent to respond to injury but only that their location is the main parameter (which was one of Reviewer1 question).

5. In many graphs the data points are so large that it is impossible to see them individually. Should be make smaller so that it doesn’t just appear as clumps of dots (Fig 1 C,D 2 E-N, 3 D,F).

Response: We agree with the reviewer and remade the graphs such that individual dots can be seen.

6. What are the cells Fig 1H that are dendra+ but don’t get photoconverted?

Response: The cells that are non-photoconverted are non-effector cells since they do not invade the neuromast or participate in phagocytosis. We clarified this point in the figure legend and in the text.

7. In the scRNAseq, find cells that are not macrophages (line 121). It is possible that the FACS was just not specific and some non-GFP+ cells got through and were sequenced. FACS gating plots are not shown. Does the scRNAseq data actually show that these other cell types express GFP and *mpeg1*?

Response: We have now added FACS plots, as well as feature plots and violin plots for *mpeg1.1* that clearly show that *mpeg1.1* expression is not specific to macrophages (new Supplementary Figure 1). We now also cite prior studies that demonstrate that *mpeg1.1* is not a specific reporter of macrophages and is expressed in other myeloid-, as well as lymphoid cells (Cavone et al., DevCell 2021; McCormack et al., Elife 2015; Ferrero et al., Journal of Leukocyte Biology 2020).

8. State in discussion (line 260) the pro-inflammatory signals “attract more macrophages to the injury site if the organ does not possess a resident population, or if the size of the injury requires more macrophages” . This isn’t really correct. First, basically all organs have a resident macrophage

population. Second, in fully developed organisms, monocytes must be recruited from the blood and these monocytes differentiate into macrophages in the tissue.

Response: We thank the reviewer for pointing out our error. Indeed, all organs have resident macrophages. We disagree that monocytes must be recruited to every injury types in adults as a general rule. Indeed, the size and type of the injury will dictate if additional monocytes are required or not. For instance while monocytes are recruited after toxic injuries in mouse muscle, it is not clear that they are present after traumatic or exercise-induced injury (see Chazaud B, Trends in immunology 2020).

9. State in discussion (line 275) that GR activation leads to chromatin unwinding and suggest that this “likely turns off the transcription of pro-inflammatory cytokines”. I don’t understand this- isn’t a permissive chromatin environment associated with increased transcription, not decreased?. Isn’t it more likely that the GR activation is inhibiting NF- κ B as many of the “pro-inflammatory” genes (ex IL1b) are activated by NF- κ B?

Response: We understand the confusion that this statement, based on the publication from Jubb et al. 2017, can generate. We decided to remove this statement to avoid more confusion.

10. Line 287: Describe results after “the loss of IL10 in zebrafish”. Should clarify there is a loss of signaling via the IL10ra receptor, not the loss of the cytokine itself in these experiments.

Response: We agree with the reviewer and have reworded this statement on page 11.

REVIEWERS' COMMENTS

Reviewer #1 (Remarks to the Author):

The authors have addressed my concerns adequately and improved the manuscript with new data.

Reviewer #2 (Remarks to the Author):

The authors have addressed all of the issues raised by the reviewers in a satisfactory manner and have added additional experimental data that strengthens the conclusions of the manuscript.

Reviewer #3 (Remarks to the Author):

In this revision, Denans et al have now added experiments demonstrating a role for IL10/IL4 signaling in synaptogenesis after hair cell injury. Additionally, the methods sections detailing the transgenic and mutant zebrafish lines have been expanded sufficiently. I only have three remaining issues that were not adequately addressed:

1. There is still confusion over the definition of “effector” macrophages. From my understanding, the confusion arises because you are defining these types of cells based on where they are at a specific early time point after injury and then measuring what happened/happens at either earlier or later time points. For example, there are “effector” macrophages inside/at the hair cells at the time of photoconversion. From my understanding, the photoconversion is used to test whether those same macrophages remain at the hair cells OR if new macrophages are recruited. The answer turns out to be the former, so that the “effector” macrophages at the time of photoconversion turn out to be the same “effector” macrophages at the time of measurement. However, this did not have to be the case. If there were newly recruited macrophages to the neuromast they would not be photoconverted but would be considered “effectors” at the time of measurement. The way it is written assumes that no new “effectors” are recruited but this did not have to be the case.

I think several further edits in the text would clarify this:

-Lines 99-101 suggest changing to: “Quantification of the location of macrophages prior to HC death shows that macrophages that will become effectors are located closer to the neuromast than macrophages that will become non-effectors”

-Figure legend lines 902-906 would suggest changing to: “Cells photoconverted (red) were inside the neuromast at the time of photoconversion while non-photoconverted cells (cyan) were not.”

-It would clarify further to draw a dotted line in Fig 1h representing the area that was photoconverted

2. Several graphs still do not allow for visualization of individual data points: Fig 2e,f,i,j,m,n; 3d,f; 4d,e

3. Is the data showing DE genes in Fig 2 d, h, l and Supp Fig 3 a, b, c, d the same as in Fig 2 c? Why is it repeated/displayed again but differently?

We thank the reviewers for their thoughtful comments throughout the review process that improved our manuscript. Please find our response to their comments below. Changes in the text are highlighted in yellow.

5 REVIEWERS' COMMENTS

Reviewer #3 (Remarks to the Author):

10 In this revision, Denans et al have now added experiments demonstrating a role for IL10/IL4 signaling in synaptogenesis after hair cell injury. Additionally, the methods sections detailing the transgenic and mutant zebrafish lines have been expanded sufficiently. I only have three remaining issues that were not adequately addressed:

15 We thank the reviewers for their thoughtful comments throughout the review process that improved our paper and for their timely review of our manuscript.

15

20 1. There is still confusion over the definition of “effector” macrophages. From my understanding, the confusion arises because you are defining these types of cells based on where they are at a specific early time point after injury and then measuring what happened/happens at either earlier or later time points. For example, there are “effector” macrophages inside/at the hair cells at the time of photoconversion. From my understanding, the photoconversion is used to test whether those same macrophages remain at the hair cells OR if new macrophages are recruited. The answer turns out to be the former, so that the “effector” macrophages at the time of photoconversion turn out to be the same “effector” macrophages at the time of measurement. However, this did not have to be the case. If there were newly recruited macrophages to the neuromast they would not be photoconverted but would be considered “effectors” at the time of measurement. The way it is written assumes that no new “effectors” are recruited but this did not have to be the case.

25

I think several further edits in the text would clarify this:

30

-Lines 99-101 suggest changing to: “Quantification of the location of macrophages prior to HC death shows that macrophages that will become effectors are located closer to the neuromast than macrophages that will become non-effectors”

35

-Figure legend lines 902-906 would suggest changing to: “Cells photoconverted (red) were inside the neuromast at the time of photoconversion while non-photoconverted cells (cyan) were not.”

40 -It would clarify further to draw a dotted line in Fig 1h representing the area that was photoconverted
We thank the reviewer for making great suggestions to make our paper more accessible. We followed their advice and modified the text accordingly.

40

45 2. Several graphs still do not allow for visualization of individual data points: Fig 2e,f,i,j,m,n; 3d,f; 4d,e
We did our best to make the data points visible, unfortunately, because of the large number of dots individual dots are not visible to the naked eye. We believe the advantage of the combined box/dot plots is that, even though not all dots are visible, it is easier to see the distribution compared to a box plot only. In addition, all the data for each graph can be found in the source data file.

3. Is the data showing DE genes in Fig 2 d, h, l and Supp Fig 3 a, b, c, d the same as in Fig 2 c? Why is it repeated/displayed again but differently?

The data in Fig 2 d, h, l and Supp Fig 3 a, b, c, d is the same as in Fig 2 c. The goal of Fig 2c was to have a

global view of DE changes, while the representation of the same data independently for each cluster allows the reader to easily read the raw values. We believe that showing this data in both ways is helpful for the reader.